# The Passage of Chaperonins to Extracellular Locations in *Legionella pneumophila* Requires a Functional Dot/Icm System

**DOI:** 10.3390/biom15010091

**Published:** 2025-01-09

**Authors:** Peter Robertson, David S. Allan, Rafael A. Garduño

**Affiliations:** 1Department of Microbiology-Immunology, Faculty of Medicine, Dalhousie University, Halifax, NS B3H 4R2, Canada; pt398201@dal.ca (P.R.); dallan@dal.ca (D.S.A.); 2Department of Medicine, Faculty of Medicine, Dalhousie University, Halifax, NS B3H 2Y9, Canada

**Keywords:** chaperonins, GroEL, HtpB, Hsp60 secretion, Dot/Icm system, immunolabeling

## Abstract

HtpB, the chaperonin of the bacterial pathogen *L. pneumophila*, is found in extracellular locations, even the cytoplasm of host cells. Although chaperonins have an essential cytoplasmic function in protein folding, HtpB exits the cytoplasm to perform extracellular virulence-related functions that support *L. pneumophila*’s lifestyle. The mechanism by which HtpB reaches extracellular locations is not currently understood. To address this experimental gap, immunoelectron microscopy, trypsin-accessibility assays, and cell fractionation were used to localize HtpB in various *L. pneumophila* secretion mutants. Dot/Icm type IV secretion mutants displayed less surface-exposed HtpB and more periplasmic HtpB than parent strains. The analysis of periplasmic extracts and outer membrane vesicles of these mutants, where HtpB co-localized with bona fide periplasmic proteins, confirmed the elevated levels of periplasmic HtpB. Genetic complementation of the mutants recovered parent strain levels of surface-exposed and periplasmic HtpB. The export of GSK-tagged HtpB into the cytoplasm of infected cells was also Dot/Icm-dependent. The translocating role of the Dot/Icm system was not specific for HtpB because GroEL, the chaperonin of *Escherichia coli*, was found at the cell surface and accumulated in the periplasm of Dot mutants when expressed in *L. pneumophila*. These findings establish that a functional Dot/Icm system is required for HtpB to reach extracellular locations, but the mechanism by which cytoplasmic HtpB reaches the periplasm remains partially unidentified.

## 1. Introduction

Gram-negative bacteria use several mechanisms to transport biomolecules from the cytoplasm to the periplasm, bacterial cell surface, extracellular milieu, or in the case of bacterial pathogens, even the interior of target (recipient) cells. Six major systems (type I to VI) have been recognized for the secretion of proteins in Gram-negative bacteria (reviewed in [1]). Type I systems rely on three gene products to accomplish protein mobilization across the inner and outer membrane in a single step [2]. On the other hand, type II, III, IV and VI secretion systems are rather complex and involve numerous proteins, including integral membrane proteins that form structural pores for the passage of secreted substrates [3,4,5,6]. The type V secretion system involves proteins known as ‘autotransporters’ [7,8], which are large multi-domain proteins that insert one of their domains in a membrane to form a pore, through which the rest of the protein gets translocated. If the translocated portion is to be released from the membrane, a catalytic domain cleaves it, leaving the membrane-inserted domain behind [9]. Other mechanisms, like the Sec-mediated pathway [10] or the twin-arginine transporter [11], may provide protein passage through the cytoplasmic membrane. Of interest is the flexibility of type IV secretion systems (T4SSs), which are capable of secreting numerous proteins, as well as mobilizing DNA [5,12].

The Dot/Icm proteins of *L. pneumophila*, the causal bacterial agent of Legionnaires’ disease in humans, are a group of more than 25 functionally related proteins [13,14,15,16,17,18] that constitute a T4SS essential for virulence. That is, Dot/Icm mutants are non-infectious, and cannot cause disease because they are unable to replicate intracellularly [14,17,19,20,21]. The Dot/Icm system also mobilizes DNA between bacterial cells [18,22], and is thus classified as a type IV-B system (T4BSS) related to the Tra plasmid-transfer system [5,12].

It has been previously reported that HtpB, the 60 kDa chaperonin of *L. pneumophila*, can be found in extra-cytoplasmic locations [23,24], and that surface-exposed HtpB mediates the invasion of HeLa cells [25] and uptake of latex beads in non-phagocytic HeLa [25] and Chinese hamster ovary (CHO) [26] cells. Moreover, when *L. pneumophila* replicates in infected HeLa cells, establishing the *Legionella*-containing vacuole (or LCV), it copiously releases HtpB into the lumen of the LCV, as has been determined by immuno-fluorescence microscopy [23] and immuno-gold electron microscopy [24]. In addition, using a C-term and N-term protein fusion of adenylate cyclase (CyaA) and HtpB as a translocation reporter, it has been demonstrated that HtpB actually reaches the cytoplasm of *L. pneumophila*-infected mammalian cells [27], where it may act as a virulence factor (reviewed in [28]). Bacterial chaperonins are typically regarded as cytoplasmic proteins that play an essential function in protein folding [29,30], and HtpB lacks the canonical molecular signals required by Gram-negative secretion systems. Therefore, it is not obvious how this protein is mobilized into the bacterial periplasm and bacterial cell surface, how it is secreted into the lumen of LCVs, or how it reaches the cytoplasm of *L. pneumophila*-infected cells. Here, we present experimental evidence that specifically implicates the Dot/Icm T4SS in the translocation of HtpB across the outer membrane, an essential step for reaching extracellular compartments (i.e., the bacterial cell surface and the cytoplasm of infected cells). In addition, we show that the Dot/Icm T4SS also mobilizes GroEL, the 60 kDa chaperonin of *Escherichia coli*, when it is expressed in *L. pneumophila* as a recombinant protein.

## 2. Materials and Methods

### 2.1. Bacterial Strains, Plasmids, and Culture Conditions

The description of *L. pneumophila* parental strains and their derivatives, as well as the *Escherichia coli* laboratory strains used are given in Table 1 [31,32,33,34,35,36,37]. In Table 2, all the plasmids used or created during this study are listed, along with their main characteristics [38,39,40,41,42]. *L. pneumophila* strains were grown at 37 °C in a humid incubator, on buffered charcoal-yeast extract plates (BCYE) [43], or in buffered yeast extract (BYE) broth [24,25], with or without supplements, as follows. *L. pneumophila* strain Lp02 and its non-virulent derivatives JV303 (*dotB^−^*), and JV309 (*dotA^−^*) were grown in media containing thymidine and streptomycin (both at 100 µg/mL). *Trans*-complemented Lp02 derivatives carrying plasmid pKB9 or pJB1153, as well as the mock-complemented Lp02 derivatives carrying plasmid pJB908, were grown on media with streptomycin (100 µg/mL) but without thymidine. *L. pneumophila* parental strains JR-32 and 130b were grown on plain BCYE plates. JR-32 derivatives GS-28K (Δ*lvh*-Kan), LELA4432 (JR-32*dotG*^−^), LELA4432-28 (*dotG*^−^Δ*lvh*), and JR-32Δ*dotA*, as well as 130b derivatives NU243 (*pilD*) and NU258 (*lspDE*), were grown on BCYE plates with 40 µg/mL kanamycin. The JR-32 derivative GS-28G (Δ*lvh*-Gm) was grown on BCYE plates with 10 µg/mL gentamicin. All JR-32 derivatives carrying pMMB, or constructs created in the pMMB backbone, were grown on BCYE plates with 5 µg/mL chloramphenicol (Cm). Legionellae stocks at −70 °C were grown once on plates before use or inoculation of a single isolated colony into BYE broth. Frozen stocks of all virulent *L. pneumophila* strains consisted of crude lysates of infected HeLa cells to which dimethyl sulfoxide (10%) was added, and maintained at −70 °C.

Stocks of non-virulent strains were made by suspending one whole isolated colony from an agar plate into 50 μL nutrient broth with 10% dimethyl sulfoxide, and kept at −70 °C.

*E. coli* strains were grown at 37 °C in LB broth, LB agar plates, or M9 synthetic medium with supplements or antibiotics as needed. JF626 (used to amplify pTrcKm) was grown on LB agar with Kan (40 µg/mL) and ampicillin (100 μg/mL). *E. coli* JM109, used to overexpress HtpB from plasmid pSH16, or GroEL from pTrcGroE, was grown at 30 °C on LB agar with ampicillin (100 μg/mL). Frozen stocks of *E. coli* strains were made by suspending one isolated colony from an agar plate into 50 μL nutrient broth with 10% dimethyl sulfoxide, and kept at −70 °C.

### 2.2. Mammalian Cell Lines and Culture Conditions

The standardized culture of the human U937 monocytic cell line (in RPMI-1640 medium) and mouse L929 fibroblastic cell line (in Minimal Essential Medium, MEM) was performed as reported previously [27]. In typical infection assays, cells were seeded either in 12-well cell culture plates at a density of 2 × 10^6^ (for U937) or 10^5^ (for L929) cells/well, or in 6-well plates at a density of 5 × 10^6^ (for U937) or 5 × 10^5^ (for L929) cells/well. Cells were typically infected at a bacteria:cell ratio of 100:1 ~16 h after seeding. The human HeLa epithelial cell line was cultured in MEM and infected with *L. pneumophila* in 25 cm^2^ cell culture flasks (Falcon Plastics-BD Biosciences Canada, Mississauga, ON, Canada), as detailed previously [25,44].

### 2.3. Molecular Biology Methods

General molecular biology techniques were performed according to the guidelines and instructions of Sambrook et al. [36], as described in Valenzuela-Valderas et al. [45]. PCR 25 μL reactions using New England Biolabs *Taq* DNA polymerase, or 50 μL reactions using Invitrogen Platinum^®^ *Pfx* DNA polymerase, were performed in a Biometra GmbH T1 thermocycler, as previously described [45]. The sequences of the primers used are given in Appendix A. All primers were commercially sourced from Integrated DNA Technologies Inc. (IDT, Coralville, IA, USA).

Due to the intrinsic resistance of *L. pneumophila* to ampicillin [46,47], a Kan^R^ marker was added to the pTrcGroE plasmid to express the GroELS proteins in *L. pneumophila*. Briefly, a genetically engineered Kan^R^ cassette was cut with *Hind*III from plasmid p34S-Km3 and ligated into plasmid pTrcGroE using the same restriction site to construct pTrcKm. Plasmids were isolated using either the Wizard^®^ Plus SV mini-prep DNA purification system (Promega, Madison, WI, USA) or the Qiagen Miniprep kit (Qiagen Inc., Cat. No. 27104, Toronto, ON, Canada), and ligation products were purified directly from agarose gels with the UltraClean^TM^ DNA purification kit (MoBio Laboratories Inc., Carlsbad, CA, USA) or the Qiagen gel purification kit (Qiagen Inc. Cat. No. 28704). Plasmid pTrcKm was electroporated into *L. pneumophila* (Lp02, and *dotA^−^* and *dotB^−^* mutants), and the expression of GroELS was induced in the exponential phase in the presence of 5 mM IPTG for 3 h. Electroporation of electrocompetent *L. pneumophila* cells was performed as described before [48].

### 2.4. Creation of Bacterial Strains and Plasmids with Specific Requirements

Strain JR-32 is more robust than strain Lp02 in infection assays, and provided better results in translocation assays performed with HtpB fused with the CyaA translocation reporter [27]. Therefore, for translocation assays with GSK-tagged proteins (see Section 2.13 below) a JR-32 *dotA* deletion mutant (JR-32Δ*dotA*) was created as an alternative to the Lp02 *dotA^−^* mutant. JR-32Δ*dotA* was created by allelic exchange using the suicide plasmid pBRDX carrying a Km^R^ cassette flanked by upstream and downstream sequences of the *dotA* gene, as depicted in Appendix A. The creation of plasmids pMGHtp, pMGLeg, and pMGMdh (encoding in-frame N-term fusions of HtpB and control proteins with the GSK translocation reporter) is depicted in Appendix A. Immunoblotting with αGSK-tag and αGSK-P confirmed that all the GSK-tagged proteins were expressed, but not phosphorylated, in *L. pneumophila*.

### 2.5. Protein Quantification

Total protein was measured using a Bio-Rad Bradford microassay (Bio-Rad Laboratories, Mississauga, ON, Canada) following the manufacturer’s instructions. The assay was run with triplicate samples in Costar 96-well flat-bottom microplates (from MilliporeSigma Canada Ltd., Oakville, ON, Canada) using a bovine serum albumin (BSA, from New England Biolabs Canada, Whitby, ON, Canada) standard curve in the range of 0–5 μg/μL. Absorbance at a wavelength of 595 nm was measured in a Benchmark Plus Microplate Spectrophotometer (Bio-Rad Laboratories, Mississauga, ON, Canada).

### 2.6. Protein Electrophoresis, Immunoblotting and Densitometry

Proteins were separated by SDS-PAGE on vertical 1.5 mm thick, 12% acrylamide mini gels in a Mini Protean II apparatus (Bio-Rad Laboratories Canada, Mississauga, ON, Canada), and electro-blotted for immunostaining following the general procedures originally outlined by Laemmli [49] and Towbin et al. [50], as previously reported [27,51]. Membranes with electroblotted proteins were pre-stained with Ponceau S (0.2% *w*/*v*, in a solution of 3% trichloroacetic acid and 3% sulfosalicylic acid in ddH_2_O) and photographed before immunostaining. Primary and secondary antibodies used for immunostaining are listed in Table 3 [52,53,54]. Some gels were stained with Coomassie Blue for general band analysis. Arbitrary bands stained with Ponceau S, as well as immunostained bands, were analyzed with GelPro software (Media Cybernetics Inc., Silver Springs, MD, USA) to determine their integrated optical density (IOD), using the “single-band analysis” function of GelPro 2.0. The IOD values were corrected for differences in loading/transfer, based on the band density data from Ponceau S-stained membranes.

### 2.7. Trypsin Assay

To determine the presence of surface-exposed chaperonins, trypsin-susceptibility assays were performed with ~10^9^ whole bacterial cells suspended in 90 μL Dulbecco-modified Eagle’s cell culture medium containing 250 µg trypsin from porcine pancreas (Gibco, now part of ThermoFisher Scientific, Waltham, MA, USA), following our previously described protocol [25]. Trypsin-accessible chaperonins were evidenced by the appearance of <60 kDa immunostained bands in Western blots.

### 2.8. Microscopy and Immunogold Labeling

Light microscopy was routinely performed using an Olympus BX6 microscope (Olympus Canada Inc., Richmond Hill, ON, Canada) equipped with a differential interference contrast (DIC) optical kit, an Evolution QEi Monochrome digital camera (Media Cibernetics, San Diego, CA, USA), and the image capture software ImagePro v.5.0 (Media Cibernetics, San Diego, CA, USA). Specimens for electron microscopy were aldehyde-fixed and processed as reported before for immunogold labeling [24,55]. Ultrathin sections mounted on nickel grids were labeled by a modified post-embedding protocol. Briefly, grids were floated for 10 min on drops of sodium borohydride (1 mg/mL) freshly dissolved in double-deionized water (ddH_2_O), and then on drops of 10 mM glycine and 100 mM Na-borate at a pH of 9. Blocking was performed for 1 h on drops of 100 mM Tris and 0.2 M NaCl at a pH of 8 (labeling buffer), containing 1% BSA and 1% skimmed milk. Then, grids were sequentially floated for 1 h on drops of the corresponding primary antibody diluted in labeling buffer containing 0.2% BSA, and then on drops of the corresponding secondary gold-conjugated antibody (Table 3) diluted in labeling buffer containing 0.2% BSA. After each antibody incubation, grids were washed 3 × 10 min in 100 mM Tris and 0.3 M NaCl at a pH of 8 (wash buffer). Finally, the labeled sections were fixed in 2.5% glutaraldehyde in wash buffer, rinsed in ddH_2_O, and stained with uranium and lead salts, before being observed in a Phillips EM 300 or a JEOL JEM 1230 transmission electron microscope equipped with a high-resolution Hamamatsu ORCA-HR digital camera. The subcellular distribution of gold particles in different compartments of a standardized bacterial cell section (typical section) was determined by a semi-quantitative analysis, following the rules reported previously [24]. Thirty random bacterial sections were analyzed per labeling experiment, and the number of HtpB epitopes (gold particles) per bacterial cell compartment were compounded as a single total, which was then reported per unit of area (for cytoplasm and periplasm) or per unit of length (for cytoplasmic and periplasmic membranes). The data were then standardized to the size of a typical section [24].

Labeling of whole intact bacterial cells (~10^7^ bacteria) was performed in suspension, specifically in 100 µL of Dulbecco-modified Eagle’s medium (DMEM) (Gibco, now ThermoFisher Scientific) containing 5 µL of αHtpB-p (Table 3). After 30 min, bacterial cells were washed once in DMEM and fixed in 4% freshly depolymerized paraformaldehyde in DMEM for 15 min. Paraformaldehyde powder was obtained from Sigma-Aldrich (Oakville, ON, Canada). The fixed bacteria were washed three times by centrifugation in DMEM with 10% newborn bovine serum (Gibco, now ThermoFisher Scientific), and then mounted on formvar-coated grids. Grids were then directly floated for 40 min on drops of αRabbitGold (Table 3) diluted in labeling buffer containing 0.2% BSA, rinsed quickly in ddH_2_O, and air-dried prior to observation without staining.

### 2.9. Osmotic Shock, Analysis of Periplasmic Proteins and Cell Fractionation

In the initial experiments with parental strain Lp02 and derivatives, periplasmic proteins were released by a modification of the osmotic shock procedure originally described by Nossal and Heppel [56], as reported by Murray et al. [57]. Briefly, bacterial cells from 100 mL BYE cultures grown to the early stationary phase (OD_620_ = 2.0 units) were pelleted (5000× *g*, 20 min), washed once with ice-cold BYE broth, and resuspended in 40 mL of 33 mM Tris-HCl at a pH of 7.1. From this point on, re-suspension in 40% sucrose and subsequent shock in 0.5 mM MgCl_2_ was performed exactly as reported by Murray et al. [57]. The final osmotic shockate (80 mL) was concentrated to a final volume of 1–3 mL using either a 10 kDa cut-off Spectra/Por type C membrane (Spectrum Medical Industries Inc., Los Angeles, CA, USA) or a 10 kDa cut-off YM-10 membrane (Millipore Co., Billerica, MA, USA) in a 60 mL Amicon concentrator (Millipore Co., Billerica, MA, USA) placed on ice and pressurized to 50 psi with N_2_ gas. When further concentration of the shockate was required, it was transferred to a Centricon YM-10 tube (Millipore cat. #42406) and centrifuged at 10^3^× *g* until the shockate was reduced to 150–250 μL. The pellet of the osmotically shocked bacteria was resuspended in 5 mL of ddH_2_0 and lysed in a Vibra-Cell sonicator (Sonics & Materials Inc., Newtown, CT, USA) in cycles of 1 min sonication, with 1 min rest on ice, for a total of 15 cycles, to produce a lysate. Corrected IOD values of immunostained bands, as well as the total protein values in lysates and shockates, were then used to estimate the % ratio of a target periplasmic protein vs. the total corresponding amount of the protein in the lysate.

In fractionation experiments with the parental strain JR-32 and derivatives expressing recombinant forms of ICDH, additional modifications to the above method were included, as follows. Bacterial cells were grown to the late exponential phase (OD_620_ = 1.5 units) and washed twice in 10 mM Tris and 33 mM NaCl at room temperature. Pelleted bacteria from the second wash were resuspended in 40 mL of 33 mM Tris. Then, equal portions of 33 mM Tris and 40% *w/v* sucrose were slowly poured into the samples with gentle agitation, resulting in a final suspension of 80 mL, comprising 33 mM Tris and 20% sucrose. Cells were then pelleted at 500× *g* for 10 min, gently re-suspended in 80 mL of ice-cold 0.5 mM MgCl_2_, and re-pelleted at 500× *g* for 10 min. The supernatant (shockate containing soluble periplasmic proteins) was collected and concentrated down to 0.5–1 mL as described above, but using a 30 kDa cut-off membrane. In parallel, the post-shockate pellet (containing mostly cytoplasmic, membrane-bound, and cell-associated proteins) was resuspended in 500 μL of 10 mM Tris and 1 mM EDTA at pH 8 (TE buffer) and sonicated using a micro-tip, with 10 s bursts followed by 30 s on ice, for 10 rounds. This lysed material was further partitioned by ultracentrifugation at 10^5^× *g* for 90 min in an Optima MAX ultracentrifuge (Beckman-Coulter, Brea, CA, USA). The supernatant (containing soluble cytoplasmic proteins) and the pellet resuspended in 1 mL TE buffer (containing membrane-associated proteins) were then collected separately.

### 2.10. Cytoplasmic Contamination of Shockates, and Efficiency of Fractionation

#### 2.10.1. Enzymatic Assays

To establish the extent of contamination of shockates with cytoplasmic proteins, glucose 6-phosphate dehydrogenase (G6PDH, EC 1.1.1.49) activity was determined by the basic method of Noltmann et al. [58] in a total volume of 3 mL. The increase in absorbance at 340 nm (A_340_) was measured in 3 mL quartz cuvettes, in a modified Cary-14 Spectrophotometer equipped with an OLIS data acquisition system (On Line Instrument Co., Bogart, GA, USA). Each cuvette contained concentrated periplasmic shockate (200–500 μg total protein), 3 μM glucose-6-phosphate (Boheringer-mannheim GmbH, now obtained through Sigma-Aldrich Canada, Co.), 30 μM MgCl_2_, and 1 μM NADP, in 0.26 mM glycyl-glycine buffer, at pH 8.0. Reagents for this assay were obtained from Sigma-Aldrich Canada Co. (Oakville, ON, Canada), unless otherwise indicated. The specific activity of G6PDH found in corresponding lysate and shockate (calculated as μmoles of NADP reduced per minute, per mg of protein), as well as the average net amount of total protein (in mg) in the lysate and shockate, were used to estimate the total G6PDH activity and relative (%) contamination of the shockate with cytoplasmic content.

In fractionation experiments with JR-32 and derivatives, alkaline phosphatase (AP) activity was used as a periplasmic marker. Active AP should be present in the periplasm, but not in the cytoplasm. Samples of different fractions (50 μg total protein) were mixed with AP buffer (100 mM Tris, 100 mM NaCl, 50 mM MgCl_2_, pH 9.5) to a final volume of 2 mL in a 3 mL spectrophotometer glass cuvette. Then, 1 mL of a 1 mg/mL solution of paranitrophenol in AP buffer was added/mixed, and the A_405_ was immediately recorded every minute for a total reaction time of 10 min.

#### 2.10.2. Immunoblotting of Iso-Citrate Dehydrogenase (ICDH)

A second method employed for estimating the cytoplasmic contamination of osmotic shockates involved detecting the cytoplasmic enzyme ICDH by immunoblotting. This method was convenient because HtpB levels were also measured by immunoblotting, and thus their levels could be more directly compared. Western blots of shockates and lysates were stained with αICDH/αRabbitAP (Table 3), and densitometry was used to determine the % contamination.

### 2.11. Detection of Bona-Fide Periplasmic Proteins

β-lactamases (periplasmic proteins used to authenticate shockates) were detected by a modification of the iodometric method of Labia & Barthélémy [59], as reported by Petroni et al. [37]. Briefly, shockates (40 μg samples) were subjected to isoelectric focusing in an IPGphor™ system, using Immobiline DryStrip IPG strips™ with a pH range of 3–10 (Amersham Pharmacia Biotech, Piscataway, NJ, USA), following the manufacturer’s instructions. Focused strips were then placed on a starch-iodine gel containing ampicillin as the developing substrate. The presence of β-lactamases was visualized by a localized clearing of the gel. A whole cell lysate of *E. coli* M3033 [37] carrying the CTX-M-2 (pI 7.9) and TEM-1 (pI 5.4) β-lactamases was also subjected to isoelectric focusing as above, and used as a positive assay control to provide pI markers.

The periplasmic *L. pneumophila* protein DsbA2 was detected by immunoblotting with αDsbA2 [52] and αMouseAP (Table 3). The detection of DsbA2 was primarily used as an efficiency marker of bacterial cell fractionation.

### 2.12. Isolation and Analysis of Outer Membrane Vesicles (OMVs)

OMVs were isolated using the method of Kadurugamuwa and Beveridge [60]. Briefly, 100 mL cultures of *L. pneumophila* strains in BYE broth (with the required supplements) were grown to the early stationary phase (OD_620 nm_ ≈ 2 units), and the bacterial cells were pelleted at 6000× *g* for 10 min. The culture supernatant was filtered through a 0.2 µm Nalgene bottle-top filter (now from MilliporeSigma, Oakville, ON, Canada), and the filtrate was centrifuged at 150,000× *g* using a Beckman 45Ti rotor in an Optima MAX ultracentrifuge (Beckman-Coulter, Brea, CA, USA). The pelleted vesicles were washed and resuspended in PBS to a final volume of 300 µL. Equal volumes of OMV suspensions were run in SDS-PAGE gels, electro-blotted onto nitrocellulose membranes, and immunostained.

### 2.13. Translocation Assays with GSK-Tagged HtpB

The GSK tag consists of a 13-amino acid peptide derived from the eukaryotic protein Glycogen Synthase Kinase [61]. This peptide is readily phosphorylated in the cytoplasm of mammalian cells, and both the phosphorylated and the non-phosphorylated forms of the peptide can be reliably detected by commercially available antibodies (Table 3). Strain JR-32, carrying either the GSK-tagged HtpB or the tagged control proteins, was grown to the stationary phase (OD_620nm_ ≥ 2) in BYE containing 5 μg/mL chloramphenicol, harvested by centrifugation at 7000× *g* for 10 min, and resuspended in cell culture medium (RPMI-1640 for U937 macrophages, or MEM for L929 cells) to a density of ≈10^10^/mL. Bacteria were then added to monolayers of cells in 12-well (1.5 mL inocula) or 6-well (3 mL inocula) cell culture plates to achieve a ratio of 600 bacteria/cell. Cell culture plates were then centrifuged at 200× *g* for 5 min, to promote contact of the inoculum with the attached cells. Plates were then incubated at 37 °C and 5% CO_2_ for 2 h in a MCO Sanyo incubator (now Panasonic Healthcare Co., Wood Dale, IL, USA). Some wells were used for immunoblotting and some for quantifying intracellular bacteria. For immunoblotting, wells were washed thrice with warm PBS, and cells were lysed in 100 μL of 2X Laemmli buffer containing a protease inhibitor cocktail and a phosphatase inhibitor cocktail (both from Sigma-Aldrich, Oakville, ON, Canada). The cell lysates were then separated by SDS-PAGE, electroblotted on nitrocellulose membranes, and immunostained with either αGSK-tag or αGSK-P in combination with αRabbitAP (Table 3). Wells used for quantification of intracellular bacteria were washed thrice with warm PBS, and then incubated for two additional hours with RPMI-1640 (for U937 cells) or MEM (for L929 cells) containing 100 µg/mL gentamicin. Cells were then washed thrice with warm PBS, and lysed in 1 mL ddH_2_O. Lysates were serially diluted in ddH_2_O and spotted on BCYE with the required supplements to determine the CFUs of intracellular legionellae per monolayer.

### 2.14. Basic Bioinformatics

The DNA/amino acid sequences of HtpB (Accession AAA25299.1), *E. coli* strain K-12 *phoA* (Gene ID: 945041), *legC6* (Gene ID: 57035579), *mdh* (Gene ID: 57036345), and *icd* (Gene ID: 57034804) were downloaded from NCBI (https://www.ncbi.nlm.nih.gov) (last accessed on 19 November 2024). The sequence of the GSK tag used to design a G-block (mini-gene) for the construction of fusions with HtpB, LegC6, and Mdh was taken from Torruelas-Garcia et al. [61].

The HtpB sequence was examined for similarities to cell penetrating peptides (CPPs) using the BLAST Search tool provided at the CPP Site 2.0: Database of Cell Penetrating Peptides (https://webs.iiitd.edu.in/raghava/cppsite/blast.php), last accessed on 19 November 2024 [62,63]. The amino acid sequence of HtpB was also visually examined for the following motifs described for Dot/Icm effectors relative to the last amino acid (−1 position) of the protein C-terminus [64,65,66,67]: (i) the presence of hydrophobic amino acids (Leu, Ile, Val, Phe) or a Pro at the −3 or the −4 position (or enrichment of hydrophobic residues in positions −1 to −3), (ii) an enrichment in negatively charged amino acids (Asp, Glu) at the −8 to −18 C−term positions, with (iii) a consequent depletion of hydrophobic amino acids at the −8 to −12 C-term positions, (iv) extensive depletion of negatively charged amino acids (Asp, Glu) at the −1 to −6 C-term positions, (v) a specific enrichment of Ser or Thr at the −3 to −11 C-term positions, and (vi) the presence of an E-block motif (usually a string of 3–5 Glu residues followed by Ile and/or Val) located within the −10 and −17 C-term positions.

### 2.15. Experiments with the C-Term of HtpB

DNA sequences encoding the last 150 or the last 300 base pairs of *htpB*, encoding the HtpB C-term 50 or 100 amino acids, were ligated in frame to the 5′ end of: (i) *icd* (*lpg0816*) lacking its stop codon, to create plasmids pICD-Cterm-50 and pICD-Cterm-100, and (ii) a G-block mini gene encoding the *E. coli* periplasmic phosphatase, PhoA, lacking a secretion signal peptide and with added restriction sites (Integrated DNA Technologies Inc., IDT, Coralville, IA, USA) to create plasmid pIP100, as depicted in Appendix A. Plasmid pIP100 was transformed by electroporation into *E. coli* strain BL-21 Δ*phoA*, which was then spotted (20 μL drops of a suspension of an OD_620_ of 1 unit) onto LB agar plates containing 60 μg/mL BCIP. Alkaline phosphatase activity was then visually detected by the formation of blue lawns.

JR32 transformed separately with pICD-Cterm-50 or pICD-Cterm-100 was fractionated to determine by immunoblotting the levels of native and recombinant ICDH present in the cytoplasm, membrane, and periplasm fractions. These fractions were tested for alkaline phosphatase activity (see above). The relative abundance of recombinant ICDH (in relation to native ICDH) was determined through calculation of the “True Secretion Ratio”, as explained in Appendix A. Because cell fractionation does not account for secreted proteins loosely associated with the outer membrane or lost in the supernatant, JR-32 cells carrying pICD-Cterm-50 were also subjected to immunogold electron microscopy with αICDH to determine the % ratios of HtpB epitopes in the compartments defined by Garduño et al. [24].

Finally, a DNA sequence encoding a string of six histidine codons in tandem was fused in-frame with the 3′ end of *htpB* lacking its stop codon. The 6-His sequence and a stop codon were part of the reverse primer P26, used to amplify *htpB* by *Pfx* PCR in combination with forward primer P25 (Appendix A). Using the restriction sites present in the primer sequences (Appendix A), this PCR amplification product was cut and subcloned into pMMB to create the plasmid pMHtp6His. The JR-32 derivative carrying plasmid pMHtp6His was processed for immunogold electron microscopy with the antibody α6XHis-tag to determine whether adding the positively charged histidine-tag to the hydrophobic C-term of HtpB would change HtpB’s compartmentalization.

## 3. Results

### 3.1. L. pneumophila T4SS Mutants Show Altered Compartmentalization Patterns for HtpB

We applied previously developed immunolocalization procedures [24] to determine the distribution patterns of HtpB epitopes in ‘typical’ ultrathin sections of *L. pneumophila* parent strains and their respective secretion mutants. Although immunogold labeling is a semi-quantitative method, it showed clear differences both in the morphology of the bacterial cell sections (Table 4), and in HtpB compartmentalization (Table 5 and Table 6). Lp02 rods were slightly thinner and shorter than those of the other strains. Additionally, both the Lp02 and JR-32 strains (including their corresponding derivatives) showed an expanded periplasm with a surface area comparable to that of the cytoplasm; that is, the separation between the cytoplasmic and outer membranes (an artifact of chemical formaldehyde fixation) was larger in relation to the reference strains SVir and 2064 (Table 4). These morphological differences may explain why the parent strains Lp02 and JR-32 showed a higher proportion of HtpB epitopes in the combined compartment of the outer membrane and the bacterial cell surface (OM-Surface) (51.5% and 65.9%, respectively) (Table 5 and Table 6), in relation to strains SVir (30.7%) and 2064 (38.8%) [24]. Consequently, HtpB epitopes in the cytoplasm + cytoplasmic membrane compartments were reduced: Lp02 (20%) and JR-32 (9.8%), versus SVir (41.8%) and 2064 (33.9%). Importantly, the proportion of HtpB epitopes in the periplasm of all strains remained fairly constant (26.87% ± 1.80%, n = 4). These observations both confirm that the amount of surface-exposed HtpB varies between strains [23,68], and emphasize the fact that immunogold labeling results should be analyzed and compared only within a given strain and derivatives.

The immunogold labeling of the Lp02 envelope followed a narrow region along the outer membrane, and often was restricted to a single row of gold particles lining the outer membrane (arrowheads in Figure 1A). In contrast, the envelopes of the *dotA*^−^ and *dotB*^−^ mutants displayed a broad labeling region along the periplasm (Figure 1A), resulting in increased % ratios of periplasmic epitopes (Table 5). Interestingly, the *dotB*^−^ mutant showed a 2.6× increase in the net number of total epitopes per section, suggesting that there was a buildup of HtpB at the bacterial cell envelope (Table 5 and Figure 1A, far right panel). The % ratio of periplasmic HtpB was even higher in the Δ*dotB* mutant, indicating a more obvious buildup, which was somewhat lessened by genetic complementation (Table 5). To clarify whether this buildup corresponded to an increase in surface-exposed HtpB, whole intact cells were immunogold-labeled in suspension. The *dotA*^−^ and *dotB*^−^ mutants did not show an increased number of surface-exposed HtpB epitopes in relation to the parent strain Lp02 (Table 7), indicating that the buildup of HtpB epitopes occurred underneath the outer membrane.

Strain JR-32 (but not Lp02) carries a second T4SS called Lvh (*Legionella* Vir homologs), which is similar to the type IV-A Vir system (T4ASS) of *Agrobacterium tumefaciens* [12,32]. Therefore, the JR-32 strain and derivatives allowed us to determine the effect of each of these T4SSs in the compartmentalization of HtpB. In relation to the Dot/Icm system, the Lvh system seemed to play a lesser role in the translocation of HtpB to extracytoplasmic compartments (Table 6). That is, the Δ*lvh*-Gm mutant lacking the Lvh system (but with a functional Dot/Icm system) showed (in relation to the parent strain JR-32) a 7% increase in HtpB epitopes associated with the cytoplasmic membrane, at the expense of a 10% reduction in epitopes at the OM-Surface, and a 0.4-fold reduction in total HtpB epitopes per typical section. On the other hand, the JR-32*dotG*^−^ mutant (with an intact Lvh system) showed a somewhat larger effect, with a 10% increase in cytoplasmic membrane labeling, a 15% reduction in epitopes at the OM-surface, and a 0.6-fold decrease in total HtpB epitopes per typical cell section. Interestingly, there was no obvious change in the % ratio of periplasmic HtpB epitopes for both mutants, in relation to the parent strain JR-32 (Table 6 and Figure 1B). Unexpectedly, the double *dotG*^−^Δ*lvh* mutant showed a unique HtpB immunolocalization pattern characterized by a 15.5% increase in internal HtpB epitopes (cytoplasm + cytoplasmic membrane labeling) at the expense of an equivalent (15.5%) decrease in envelope-associated epitopes (periplasm + OM-Surface compartments). However, it should be noted that at the individual level, there were some bacterial cell sections of the double mutant that clearly showed a visual increase in periplasmic labeling, similar to that observed in the Lp02 *dot* mutants (e.g., far right micrograph in Figure 1B). The combined facts that the *dotG*^−^ mutant did not show a buildup of periplasmic HtpB, and that the double T4SS mutant had a unique distribution of HtpB epitopes, suggested that DotG is not essential for HtpB translocation, and confirmed a functional interaction of the Dot/Icm and Lvh T4SSs [32].

The possible functional interactions between secretion systems [69], in combination with the accumulation of HtpB in the periplasm of *dot* mutants, further suggested that secretion mechanisms other than the Dot/Icm system could mediate the passage of HtpB to the periplasm/cell surface. Thus, type II secretion mutants of strain 130b (Δ*pilD* and Δ*lspDE*) were immunogold−labeled, but it was found that they did not show obvious differences in the distribution of HtpB epitopes in relation to the parent strain.

### 3.2. Dot/Icm Mutants Display a Reduced Amount of Surface-Exposed (Trypsin-Accessible) HtpB

Trypsin assays are mostly qualitative (providing a “+” or “−” result), or at times semi-quantitative (providing a “<” or “>” result) in relation to the HtpB displayed on the surface of intact bacterial cells. The amount or % ratio of surface-exposed HtpB cannot be quantified in this assay. Trypsin treatment of whole cells of the virulent parent strain Lp02 generated a number of HtpB degradation products (visible as αHtpB-m immunostained bands in Figure 2A), indicating that a fraction of the total HtpB was accessible to trypsin either on the surface of Lp02 cells or in the assay supernatant. The *dotA*^−^, *dotB*^−^, and Δ*dotB* mutants showed a distinct band of full-length HtpB, but virtually no degradation products (Figure 2), indicating that they had less surface-exposed or released HtpB than strain Lp02. The specific involvement of a functional Dot/Icm system in the localization of HtpB to the cell surface was demonstrated by genetic complementation. That is, the *trans*-complemented strains *dotA*^−^ + C, *dotB*^−^ + C, and Δ*dotB* + C (but not the mock-complemented strains *dotB*^−^ + V and Δ*dotB* + V) showed a recovery of trypsin-accessible HtpB (Figure 2).

The parent strain JR-32 and its derivative mutants Δ*lvh*-Kan and Δ*lvh*-Gm (with deletions in the Lvh system, but an intact Dot/Icm system) showed clear degradation products from trypsin-accessible HtpB (Figure 3A). The JR-32*dotG*^−^ mutant (with a non-functional Dot/Icm system but an intact Lvh system) showed a slight decrease in trypsin-accessible HtpB, which did not decrease further in the *dotG*^−^Δ*lvh* double mutant (defective in T4S altogether) (Figure 3A). Thus, while immunogold labeling assays indicated that the Lvh system affected the distribution of HtpB epitopes in internal compartments, trypsin assays showed that it is not involved in the surface localization of HtpB. The fact that the level of trypsin-accessible HtpB in JR-32*dotG*^−^ was not reduced to the same extent as observed in Lp02 *dot* mutants suggested once more that DotG is not essential for HtpB translocation, or that it might be partially compensated through a functional interaction with components of the Lvh system.

In trypsin assays performed with the Δ*pilD* and Δ*lspDE* T2SS mutants, they appeared to display more trypsin-accessible HtpB than the 130b parent strain, not only because there were more obvious HtpB degradation products, but also because the band corresponding to the full-size HtpB was almost gone (Figure 3B). Interestingly, the trypsin-free controls of these mutants also showed a clear degradation of HtpB, suggesting the existence of an internal proteolytic process, independent from the addition of exogenous trypsin. A Coomassie Blue-stained SDS-PAGE sister gel showed that many protein bands in the higher molecular weight range had in fact been degraded (Figure 3C). This was an unexpected result, likely linked to the previously reported changes in the outer membrane properties of these T2S mutants [35,70], as well as the backlog of T2S substrates. Therefore, we decided not to use these strains for further experimentation, even though any potential role of the *L. pneumophila* T2SS in HtpB secretion remained unresolved.

### 3.3. Analysis of Osmotic Shockates Confirmed the Increased Periplasmic Levels of HtpB in Dot Mutants

It was hypothesized that if HtpB transiently resides in the periplasm of parent strains and builds up in *dot* mutants, it should be possible to pick it up and quantify it in periplasmic ‘shockates’. To account for contamination of shockates with cytoplasmic content, the activity of glucose-6-phosphate dehydrogenase (G6PDH) was measured in both cell lysates and shockates. Because G6PDH should not be present in the periplasm, the % contamination was calculated as 100 × (total periplasmic G6PDH activity/total G6PDH activity in the corresponding lysate). The % contamination was then subtracted from the % ratio of total periplasmic HtpB/total HtpB in the lysate (measured by densitometry) to determine the % of true periplasmic HtpB (Table 8).

HtpB was present in the shockates of all samples, but the amount of true periplasmic HtpB found in the *dot* mutants was consistently higher than that in the parent strain Lp02 (Figure 4 and Table 8). Importantly, *trans*-complementation of the *dotA*^−^, *dotB*^−^, and Δ*dotB* mutants caused a reduction in the proportion of HtpB found in the concentrated shockates, indicating that the Dot/Icm system is sufficient in the mobilization of periplasmic HtpB. The only departure from the expected results was the mock complementation control (*dotB*^−^ + V strain) in the second fractionation experiment, which showed a true periplasmic HtpB % value that was more similar to that of the parent strain, than to the *dotB*^−^ mutant (Table 8).

Next, it was determined whether shockates of strain Lp02 and the *dotB*^−^ mutant would contain a known *L. pneumophila* periplasmic protein. Philadelphia-1 strains (like Lp02) produce a β-lactamase (a class 2d oxacillinase) with a pI above 8.0 [47,71] that resides in the periplasm [46]. Strong β-lactamase activity was detected at the correct position (pI > 8.0) in a periplasmic shockate from the parent strain Lp02 (Figure 5). Furthermore, periplasmic shockates from Lp02 and the *dotB*^−^ mutant harboring the plasmid pTrcKm, which carries the *bla* gene (Amp^R^ marker) encoding the β-lactamase TEM-1 [72], showed massive lactamase activity at the pI = 5.4 position (Figure 5). These results confirmed that osmotic shockates contained a bona fide periplasmic *L. pneumophila* protein, and that the recombinant β-lactamase expressed from plasmid pTrcKm was properly fractionated. Importantly, the levels of periplasmic oxacillinase were not augmented in the *dotB*^−^ mutant, suggesting that the DotB defect did not cause a general accumulation of periplasmic proteins.

A second method to monitor the contamination of periplasmic shockates was the detection of the cytoplasmic enzyme iso-citrate dehydrogenase (ICDH) by immunoblotting, which allowed a more direct comparison between ICDH and HtpB levels (measured by the same methodology). Any ICDH present in shockates was considered to originate in the cytoplasm, hence the ratio of (total shockate ICDH/total lysate ICDH) × 100 was used as the % contamination. In general, this method resulted in small (or negative) % ratios of true periplasmic HtpB (e.g., Table 9, experiment 1), and in one experiment, the % values of true periplasmic HtpB were negative for all (Lp02 and *dotB*^−^) samples. Negative values were interpreted as 0% true periplasmic HtpB, meaning that all the HtpB present in the concentrated shockate originated from cytoplasmic contamination. To determine whether a bona fide periplasmic protein would also yield small % periplasmic proportions via this method, the detection of DsbA2 by immunoblotting was included in two additional fractionation experiments. Interestingly, the % values of true periplasmic DsbA2 turned out to be low as well (Table 9), suggesting that these low % ratios were intrinsic to the immunoblot methodology. Nonetheless, the % ratio of true periplasmic HtpB (albeit small) was higher in the *dotB*^−^ mutant than in the parent strain Lp02, and trans-complementation of the *dotB*^−^ mutant (but not the mock-complementation) caused a reduction in periplasmic HtpB (Table 9).

### 3.4. HtpB Co-Localizes with DsbA2, a Bona Fide L. pneumophila Periplasmic Protein, in Outer Membrane Vesicles (OMVs)

Gram-negative bacteria produce OMVs that are released into culture supernatants (reviewed in [73]). When they form, these OMVs naturally entrap periplasmic proteins [74,75]; hence, periplasmic HtpB should be entrapped in OMVs. In fact, HtpB has been previously identified in a proteomic analysis of *L. pneumophila* OMVs [76].

HtpB was detected by immunoblotting in OMVs isolated from Lp02 culture supernatants (Figure 6A), and as expected, in equivalent OMV samples, there was more HtpB in the *dotA*^−^ and *dotB*^−^ mutants than in Lp02. OMVs of *trans*-complemented *dotA*^−^ (Figure 6A) or *dotB*^−^ mutants had undetectable levels of HtpB, specifically implicating (once more) the Dot/Icm system as sufficient in the mobilization of periplasmic HtpB. Western blots of OMVs immunostained for location markers showed both that HtpB co-localized with the periplasmic protein DsbA2 and OMVs did not contain the cytoplasmic enzyme ICDH (Figure 6B), suggesting that the HtpB detected in OMVs was truly periplasmic. Interestingly, the monoclonal antibody αHtpB-m (which recognizes HtpB’s C-term) [41] was unable to immunostain the HtpB contained in OMVs, an observation confirmed in further labeling experiments. It seems, thus, that the HtpB contained in OMVs has its C-term either hidden (through structural changes or interactions with lipids), or possibly cleaved off. The αHtpB-p immunostained band running at the top of the 46 kDa marker (clearly present in Lp02 OMVs, but barely seen in the *dotB*^−^ mutant) (Figure 6B, far left) could represent such a cleaved form of HtpB. However, the fact that there is still a detectable amount of full size HtpB in these samples, which was not picked up with αHtpB-m, favors the notion that the C-term is hidden.

### 3.5. The Translocation of HtpB into the Cytoplasm of Host Cells Requires a Functional Dot/Icm System

To test whether or not the translocation of HtpB to the cytoplasm of mammalian host cells requires a functional Dot/Icm system, GSK-tagged derivatives of JR-32 and JR-32Δ*dotA* were used to infect U937-derived human macrophages. The three tagged proteins were detected in lysates of infected cells immunostained with αGSK-tag, indicating that the recombinant proteins were expressed in host cells (Figure 7). However, the detection of GSK-LegC6 (positive translocation control) was obscured by adjacent cross-reactive protein bands from U937 cells (Figure 7A). Immunostaining with the αGSK-P antibody only exposed a faint band of phosphorylated GSK-LegC6 (Figure 7B).

The detection of both phosphorylated GSK-HtpB and GSK-Mdh was severely obscured by cross-reactive shadow bands similar in size to the targets.

The experiment was thus repeated with L929 mouse fibroblast cells, which have been previously used in successful infection experiments with *L. pneumophila* [27]. The three tagged proteins were clearly detected in lysates of infected L929 cells immunostained with αGSK-tag because of a cleaner labeling background in relation to U937 cells (Figure 7C). Additionally, in this case, a band of phosphorylated GSK-HtpB was clearly detected in cells infected with the parent strain JR-32, but not with the JR-32Δ*dotA* mutant, indicating that the translocation of HtpB into the cytoplasm of infected cells requires a functional Dot/Icm system. The translocation controls worked as expected: (i) phosphorylated GSK-LegC6 (positive translocation control) was detected in cells infected with the parent strain, but not in cells infected with the JR-32Δ*dotA* mutant (Figure 7D), and (ii) the phosphorylated GSK-Mdh (negative translocation control) was neither detected in cells infected with the parent strain JR-32, nor with the JR-32Δ*dotA* mutant. The mean number of intracellular legionellae was very similar for the six strains involved in these experiments (JR-32 and JR-32Δ*dotA*, each expressing the three tagged proteins) at 1.73 ± 0.67 × 10^6^ (n = 12) per monolayer.

### 3.6. HtpB Does Not Have All the C-Term Amino Acid Motifs of Dot/Icm Effectors

The mechanism by which T4SSs secrete their substrates is still not fully understood, but Dot/Icm effectors show a set of common amino acid motifs in their C-terms that could be used as predictors for secretion. Although HtpB seemingly acted as a substrate of the Dot/Icm system in the above translocation assay, it only met one of the six predictors fully, and three of them partially (Figure 8). Both the −3 and −4 C-term residues in HtpB are glycine, a somewhat neutral amino acid (−0.4 on the Kyte-Doolittle scale), meaning that HtpB does not meet the criterion of having a hydrophobic amino acid or a Pro in those positions. However, considering methionine as a moderately hydrophobic amino acid (+1.9 on the Kyte-Doolittle scale), its presence in positions −1, −2 and −5 could partially meet the criterion. With a single Asp in position −15, HtpB cannot be considered enriched in negatively charged amino acids in the −8 to −18 C-term positions. Again, considering methionine as a moderately hydrophobic amino acid, HtpB is not totally depleted of hydrophobic residues in positions −8 to −12, and thus only meets this criterion partially. HtpB fully met the requirement of being depleted of Glu or Asp at the −1 to −6 C-term positions, but did not show any Ser or Thr residues between the −3 to −11 C-term positions. Finally, HtpB showed a short E-block motif consisting of two glutamic acid residues followed by a valine in the −19 to −22 positions, which is outside of the canonical −10 to −17 positions (Figure 8). Although not strong enough to support the notion that HtpB is an authentic Dot/Icm T4SS substrate, the C-term features of HtpB suggest that it could play a role in HtpB translocation.

### 3.7. The C-Term of HtpB Proved to Play a Role in Mediating Membrane-Association and Potential Translocation of HtpB Across the Cytoplasmic Membrane of L. pneumophila

The fractionation of JR-32 derivatives carrying either plasmid pICD-Cterm-50 or pICD-Cterm-100 indicated that the recombinant ICDH proteins were preferentially located in the membrane fraction (Figure 9). The True Secretion Ratio (TSR) method used in the analysis of these fractionation experiments (Appendix A) is fundamentally different from the approach used above to evaluate osmotic shockates versus lysates, which was based on % ratios of periplasmic HtpB corrected by the % contamination. The TSR was used to determine the relative abundance of recombinant ICDH forms in a given cellular fraction, in relation to the abundance of native ICDH in the same fraction. That is, because all the ICDH forms were detected by the same antibody in the same sample lane of the immunoblot, native ICDH worked as an internal localization/contamination control for each fraction. A TSR value of zero would indicate that the proportion of a recombinant ICDH form in a given fraction is exactly the same as the proportion of native ICDH (ruling out a legitimate mobilization). A TSR value of 1 would indicate that a recombinant ICDH form was present in a fraction where the native ICDH was absent (validating a legitimate translocation). A plot of the TSR results from three independent experimental replicas is shown in Figure 9C. Because the TSR values of the two recombinant ICDH forms for the membrane fraction were close to the value of 1, it was inferred that the C-term of HtpB truly mediated the association of recombinant ICDH forms with *L. pneumophila* membranes. The plot also showed that no recombinant ICDH forms (or only a very small portion) legitimately passed into the periplasm of *L. pneumophila* (Figure 9C). It should be noted that the cytoplasmic fraction of these experiments was clean, showing no alkaline phosphatase activity.

Immunogold labeling with αICDH (which recognizes both the native and the recombinant forms of ICDH) showed that in ultrathin sections of control JR-32 cells carrying the empty vector pMMB, 12.4% of the native ICDH epitopes were present in the combined compartments of cell surface + outer membrane + periplasm, whereas 87.6% were present in the combined compartments of cytoplasm + inner membrane [77]. On the other hand, in sections of JR-32 cells carrying pICD-Cterm-50, 22.9% of the gold particles were present in the cell surface + outer membrane + periplasm compartments, and 77.1% were present in the cytoplasm + cytoplasmic membrane compartments [77]. Collectively interpreting these results, we concluded that a portion of the recombinant ICDH-50 that partitioned with membranes (as shown in the fractionation experiment), had gone, according to immunogold labeling, past the inner membrane to associate with the outer membrane.

Additional support for a role of the C-term in HtpB translocation came from an immunogold labeling experiment with JR-32 expressing HtpB-6His, in which 77.0% of HtpB epitopes were present in the cytoplasm + cytoplasmic membrane compartments, whereas only 8.1% were in the periplasm and 14.9% in the OM surface [77]. Although these results (obtained with antibody α6XHis-tag) cannot be directly compared to the JR-32 results in Table 6 (obtained with αHtpB-p), the extensive change in HtpB compartmentalization indicated that the 6His-tag hindered the passage of HtpB to extra-cytoplasmic locations.

Finally, it was determined that the C-term of HtpB could not mediate the passage of alkaline phosphatase (PhoA) across the cytoplasmic membrane of *E. coli* into the periplasm. PhoA is activated in the periplasm and can be visually detected using chromogenic substrates (like BCIP). While the *E. coli* BL-21 parent strain developed a characteristic blue color on LB agar containing BCIP, neither the BL-21 Δ*phoA* strain nor the Δ*phoA* strain carrying pIP100 developed color [77]. To rule out the possibility that a component of the LB medium could influence the development of blue color, the test was repeated on M9 medium, but the result was the same.

### 3.8. The Dot/Icm System Mobilizes Recombinant GroEL Expressed in L. pneumophila

GroEL, the *E. coli* 60 kDa chaperonin (or Hsp60), is a typical cytoplasmic protein, even though under certain stress conditions it associates with membranes (via its C-term) to act as a lipochaperonin [78]. Recombinant HtpB expressed in *E. coli* largely remains in the cytoplasm [24], suggesting that *E. coli* lacks the mechanisms used by *L. pneumophila* to mobilize HtpB to extra-cytoplasmic locations. Thus, GroEL was expressed in *L. pneumophila*, and its location was assessed by trypsin-accessibility assays and cellular fractionation. Immunogold labeling could not be used here because labeling with monoclonal αGroEL was only marginally better than the background labeling observed for the negative (no primary antibody) control.

*L. pneumophila* Lp02, but not the *dotA*^−^ or *dotB*^−^ mutants, displayed trypsin-accessible recombinant GroEL (Figure 10B,C). The analysis of periplasmic shockates showed that 2.6% of the total recombinant GroEL expressed in Lp02 was periplasmic (Figure 10E and Table 10), and that this amount increased to 3.9% and 5.1% in the *dotA*^−^ and *dotB*^−^ mutants, respectively (Figure 10E and Table 10). GroEL was also detected by immunoblot in OMVs isolated from IPTG-induced Lp02 + pTrcKm. We concluded that GroEL both is secretion-competent in *L. pneumophila* and follows HtpB’s fate in the *dot* mutants.

In a control experiment, the absence of both trypsin-accessible Hsp60 (Figure 11A,B) and periplasmic HtpB (Figure 11C and Table 10) was confirmed in *E. coli* JM109 either expressing HtpB or over-expressing GroEL. However, some of the over-expressed GroEL was trypsin-accessible (Figure 11A), and 0.51% of it was found in the periplasm of JM109 + pTrcGroE (Figure 11D and Table 10), suggesting that high levels of GroEL may partially overcome a bottleneck that controls its passage past the *E. coli* cytoplasmic membrane. Interestingly, the recombinant HtpB (but not the over-expressed GroEL) was somewhat degraded in *E. coli*, as seen in the control lane for JM109 + pSH16 in Figure 11B. These results confirm that *L. pneumophila*, but not *E. coli*, possesses the mechanisms required to mobilize Hsp60 to extra-cytoplasmic and extracellular locations.

## 4. Discussion

The experimental work presented here was meant to elicit some understanding of the mechanism by which HtpB is translocated across the *L. pneumophila* envelope. A combination of non-quantitative and semi-quantitative methods was used to map the presence of HtpB in particular cell compartments, including the cytoplasm of host cells. The co-localization of periplasmic HtpB with bona fide periplasmic proteins (Figure 4, Figure 5 and Figure 6 and Table 5, Table 6, Table 8 and Table 9) and its presence in the cytoplasm of host cells (Figure 7) were unequivocally confirmed. More importantly, it was demonstrated that the presence of HtpB on the bacterial cell surface and in the cytoplasm of host cells requires a functional Dot/Icm T4SS. These are two key extracellular locations where HtpB performs virulence-related functions [28,79], so it seems reasonable to hypothesize that the translocation of HtpB co-evolved with its virulence-related functions.

Although the translocation of HtpB was not fully expounded, HtpB’s buildup in the periplasm and the parallel decrease of surface-exposed HtpB in the *dot* mutants (Figure 1 and Figure 2) clearly pointed at a stepwise mechanism. The passage of HtpB from the cytoplasm to the periplasm would be a separate step from the subsequent mobilization of HtpB to the bacterial cell surface (a step interrupted in the *dot* mutants). The observed relief of the periplasmic backlog and the restoration of surface-exposed HtpB in *trans*-complemented mutants (Table 5, Figure 2), shows that the Dot/Icm system directly mobilizes HtpB across the outer membrane, implying that HtpB engages the Dot/Icm system in the periplasm (see below).

For *L. pneumophila* cells in the LCV, the extracellular milieu is the LCV’s lumen, and HtpB was confirmed to reach the host cell cytoplasm in a Dot/Icm-dependent manner. Thus, another step in the transport of HtpB would be the crossing of the LCV membrane, in which the Dot/Icm system could play a direct or indirect role. A direct role would mean that, in vivo, the step of crossing the outer membrane is coupled with the crossing of the LCV membrane, denoting that HtpB is actively ‘injected’ like a Dot/Icm substrate. Since HtpB accumulates in the lumen of LCVs [23,24], a direct injection into the host cell cytoplasm can be convincingly ruled out. However, a partially direct role could still be envisioned, in which the HtpB that accumulates in the LCV lumen crosses the LCV membrane through pores formed by Dot/Icm-secreted effectors. Evidence for the existence of such pores comes from both the observed Dot/Icm-dependent pore-forming activity of *L. pneumophila* [80] and the tight contact points observed between *L. pneumophila*’s outer membrane and the LCV membrane [81], recently shown to involve structural components of the Dot/Icm secretion apparatus [82]. DotA is a membrane protein [83], proposed to form a ring structure within the inner membrane complex of the Dot/Icm secretion apparatus [84]. Additionally, DotA is secreted by the Dot/Icm system, and has been proposed to form pores in the LCV membrane [85]. In fact, the DotA homolog of *Coxiella burnetii* has been detected on the membrane of the parasitophorous vacuole (the equivalent of the LCV) in infected HeLa, RK13, and Vero cells [86]. Therefore, it seems reasonable to hypothesize that DotA could form pores in the LCV membrane through which HtpB could pass into the host cell cytoplasm.

An indirect Dot/Icm role in the passage of HtpB across the LCV membrane would mean that this system is just required to transport HtpB across the outer membrane, leaving it in the lumen of the LCV, from where HtpB (on its own) would cross the LCV membrane. This process would rely on HtpB’s ability to interact with, penetrate, and cross the membrane. An analysis of the amino acid sequence of HtpB showed that it displays similarities with cell penetrating peptides (CPPs). CPPs are short amino acid sequences (<40 residues long) capable of wiggling their way across lipid bilayers and are naturally abundant among venoms and toxins (reviewed in [87]). Remarkably, CPPs can move a variety of molecular cargoes across membranes, including covalent and non-covalent complexes. HtpB amino acids 168–173 showed similarity to a short internal sequence of the antimicrobial peptide LL-37 [88], amino acids 321–345 showed similarity to CPP Inv3.7 derived from the MceA1 protein of *Mycobacterium tuberculosis* [89], and amino acids 442–454 showed similarity to the synthetic CPP S6R [90]. The similarity to Inv3.7 is particularly relevant, because Inv3.7 mediates the translocation of MceA1 into the cytoplasm of host cells [89].

Regarding the passage of HtpB from the bacterial cytoplasm to the periplasm, it could be ostensibly concluded that this is a Dot/Icm-independent step that still happens in *dot* mutants. However, as reported for other T4SSs [5], the Dot/Icm system comprises a type IV coupling complex (T4CC) that assembles in the cytoplasmic membrane (reviewed in [91,92]). T4CCs play an essential role in substrate recognition and sorting, engaging cytoplasmic substrates and directing them to the secretion apparatus. The Dot/Icm T4CC, in particular, displays two modes of substrate recruitment. In the first mode, substrates bind to the chaperone-like adaptor proteins IcmS, IcmW, and LvgA, and are directed to the secretion apparatus. The second mode is a two-step mechanism that involves a direct interaction of substrates with DotL, a primary component of the T4CC. Importantly, the interaction of substrates with DotL requires the C-term E-block motif (which HtpB partially meets, Figure 8). The interaction with DotL leads to the translocation of substrates to the periplasm (step 1), from where they engage (via an unknown mechanism) the Dot/Icm secretion apparatus (step 2) [5,91,92]. Two-step secretion constituted the basis of two of the three mechanistic models of type IV secretion proposed in an early review [93], and periplasmic intermediates of T4SS substrates have been known for some time. Pertussis toxin subunits are first transported to the periplasm of *Bordetella pertussis* in a Sec-dependent manner, and are then secreted by the Ptl T4SS [94]. Further, some T4ASS substrates in *A. tumefaciens* are translocated to the periplasm before being injected into the plant cell by the VirB T4ASS [95]. Therefore, if HtpB directly interacts with DotL, its passage to the periplasm would be T4CC-mediated.

DotB is an hexameric cytoplasmic ATPase that energizes the Dot/Icm system [91,92]. DotB hexamers transiently interact with DotO hexamers at the base of the secretion apparatus, inserted in the inner membrane. When DotB binds to DotO, it causes conformational changes, not only in DotO, but also in the whole secretion apparatus [96]. The Dot/Icm T4CC interacts with the secretion apparatus only in the first mode of substrate recruitment; in the second mode, it acts independently. Consequently, mutations in *dotA* or *dotB* would have no effect on the second mode function of the T4CC, thereby explaining why it would be possible for *dotA*^−^, *dotB*^−^, and Δ*dotB* mutants to still accumulate HtpB in the periplasm. Phenotypical differences between the *dotB*^−^ and the Δ*dotB* mutants could be explained by the fact that in the *dotB*^−^ mutant, a non-functional DotB would still be present and able to bind DotO [96], whereas in the Δ*dotB* mutant, no DotB hexamers would be present, and the structure of the secretion apparatus would thus be locked in a closed conformation. The T4CC would then be permanently disengaged from the secretion apparatus, and free to translocate substrates to the periplasm, thereby explaining the more obvious backlog of periplasmic HtpB in the Δ*dotB* mutant (Table 5).

A Dot/Icm-independent scenario for the passage of HtpB across the cytoplasmic membrane would either be the result of HtpB’s ability to interact with and penetrate membranes, involve the interaction of HtpB with a membrane translocator different from the T4CC, or both. In relation to HtpB’s ability to interact with the cytoplasmic membrane, it is remarkable that in the early years of *L. pneumophila* research, HtpB was named the “major cytoplasmic membrane protein”, highlighting its ability to partition with the cytoplasmic membrane [97,98]. When the HtpB C-term was fused to the cytoplasmic enzyme ICDH, the recombinant ICDH partitioned with the membrane fraction (Figure 9), and adding a 6His-tag to the HtpB C-term, caused a reduction in extra-cytoplasmic HtpB epitopes, collectively suggesting that association with membranes favors passage to extra-cytoplasmic locations. However, it is unlikely that the ability to associate with membranes would be sufficient for HtpB to cross the bacterial cytoplasmic membrane on its own. If that was the case, recombinant HtpB expressed in *E. coli* should have managed to reach the periplasm, but that was not observed (Figure 11).

On the contrary, the experiments with recombinant GroEL in *L. pneumophila* (Figure 10), in combination with the failure of HtpB C-term to mediate the passage of PhoA into the *E. coli* periplasm, indicated that a translocator present in *L. pneumophila* (but absent in *E. coli*) was required for the passage of GroEL (and HtpB) to extra-cytoplasmic locations. However, it is still possible that association with membranes could enhance the interaction with this proposed translocator. The potential involvement of alternate translocators (different from the Dot/Icm system) was partially explored with the JR-32 Lvh mutants and the T2SS mutants (Figure 3 and Table 6). The JR-32 Δ*lvh*-Gm mutant showed an increase in HtpB epitopes in the cytoplasmic membrane, but did not show an equivalent reduction in periplasmic epitopes. Additionally, the Lvh mutants did not show any reduction in trypsin-accessible HtpB, denoting that Lvh (*per se*) did not play a role in the passage of HtpB into the periplasm. Secondarily, these experiments pointed at peculiarities of the JR-32*dotG*^−^ mutant, which did not behave like the Lp02 *dotA* and *dotB* mutants. DotG is a homolog of Lvh’s VirB10, and forms part of the core complex of the Dot/Icm secretion apparatus [99]. The JR-32*dotG*^−^ mutant still mobilized HtpB to the cell surface (Figure 3), indicating that DotG is dispensable for HtpB translocation, which correlates well with the fact that DotG is also dispensable for the assembly of the Dot/Icm core complex [91,92]. In addition, although Lvh’s VirB10 could complement the function of DotG, this possibility was ruled out because it has been previously reported that strain JV573, an Lp02 *dotG*_Δ431–935_ mutant that does not carry the Lvh system [80], also displayed trypsin-accessible HtpB [100]. The JR-32*dotG*^−^ mutant, which carries a transposon insertion (Table 1), and strain JV573 both produce truncated DotG proteins. The truncated DotG of strain JV573 has been proposed to hinder the secretion of Dot/Icm substrates [80,101], while the absence of DotG in a clean in-frame Δ*dotG* mutant (JV3559) appeared to still have an open channel, making JV3559 hyperhemolytic [101]. We predict that strain JV3559 would have parent strain levels of surface-exposed HtpB, mainly because this strain is not defective in HeLa cell invasion [101], a trait that depends on surface-exposed HtpB [25]. However, this would have to be confirmed experimentally. Thus, the choice of a *dotG* mutant of strain JR-32 could not fully resolve the role of the Lvh system in the translocation of HtpB. As will be discussed below, a JR-32 Δ*dotA* or Δ*dotL* mutant could be a more definitive choice for this purpose.

While the overall results from experiments performed with the Δ*lspDE* and Δ*pilD* T2SS mutants were inconclusive, immunogold labeling in particular suggested that the T2SS of *L. pneumophila* was not involved in the compartmentalization of HtpB. Furthermore, a role of the *L. pneumophila* T1SS [102,103] or the flagellar T3SS [104] was ruled out because periplasmic substrate intermediates have not been described for these systems [2,4,105]. Remaining candidates that could translocate HtpB to the periplasm are the Sec-mediated pathway, or the *L. pneumophila* twin arginine transporter (TAT) [106]. These two systems require well-defined secretion signal sequences that HtpB or GroEL do not possess. Although these chaperonins could potentially hitchhike on TAT substrates [107] or interact with Sec proteins [108], both transporters could be ruled out because the potential translocator that moves HtpB to the periplasm should not be present in *E. coli*, and *E. coli* has both a Sec-mediated pathway [108] and a TAT system [109]. Therefore, a potential translocator of HtpB (and GroEL) different from the T4CC could not be proposed at this point.

Many questions regarding the secretion mechanism of HtpB were not answered, but now, guiding directions remain as to what future experiments could address these gaps. For instance, the analysis of a Δ*dotL* mutant could clarify whether or not the Dot/Icm T4CC mediates the passage of HtpB from the cytoplasm to the periplasm. The analysis of the JR-32Δ*dotA* mutant and its *trans*-complemented derivative should also prove to be a better alternative to the analysis completed here with the JR-32 Δ*dotG* mutant, to render clearer results on the role of the Lvh system in HtpB mobilization. Testing the JR-32 strain expressing the 6His-tagged HtpB, using the α6XHis-tag antibody throughout, should also provide answers regarding the compartmentalization role played by the HtpB C-term. In relation to the potential ability of HtpB to cross the LCV membrane, translocation experiments with GSK-tagged recombinant HtpB forms (either carrying the C-term 6His tag, or site-directed mutations in the regions that show similarity to CPPs) should provide definitive answers, not to mention the addition of an exciting experiment involving a GSK-tagged GroEL expressed in *L. pneumophila*.

## 5. Conclusions

We have shown that the mobilization of HtpB to extracellular locations in *L. pneumophila* is a multistep process in which the Dot/Icm system is involved. Step 1 would be the Dot/Icm-independent association of HtpB with the cytoplasmic membrane. Although not sufficient for translocation, this association could favor the interaction of HtpB with a *L. pneumophila* translocator that would mediate Step 2 of the process: the crossing of the inner membrane. This putative translocator could either be the Dot/Icm T4CC, or a yet to be discovered one. Step 3 would be the crossing of the outer membrane, which was demonstrated here to be a Dot/Icm-mediated process that requires a direct interaction with periplasmic HtpB. The outcome of Step 3 would be the display of surface-exposed HtpB in vitro, or the accumulation of HtpB in the lumen of the LCV in vivo. Step 4 would be the crossing of the LCV membrane to place HtpB in the cytoplasm of the host cell. This final step could be achieved either through pores produced by a functional Dot/Icm secretion apparatus, or through the ability of HtpB to interact with membranes. The unequivocal direct involvement of the Dot/Icm T4BSS in step 3 of the process constitutes the major contribution derived from the work presented here.

## Figures and Tables

**Figure 1 biomolecules-15-00091-f001:**
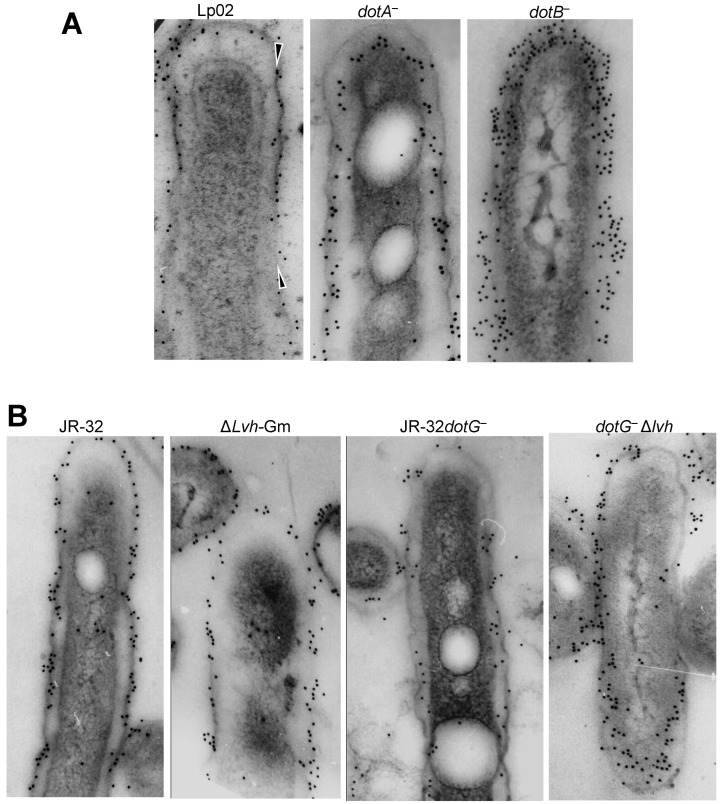
Electron micrographs of ultrathin sections of the *L. pneumophila* parent strains and mutants (indicated on top of each panel), immunolabeled with HtpB-specific hyperimmune serum and anti-rabbit goat IgG conjugated with gold spheres. (**A**) Side-by-side comparison of sectioned Lp02, *dotA^−^* mutant, and *dotB^−^* mutant. Arrowheads point at a row of gold particles lining the outer membrane of Lp02. (**B**) Side-by-side comparison of sectioned JR-32, Δ*lvh*-Gm mutant, JR-32*dotG*^−^ mutant, and *dotG*^−^Δ*lvh* double mutant. For size reference, gold spheres are ~10 nm in diameter.

**Figure 2 biomolecules-15-00091-f002:**
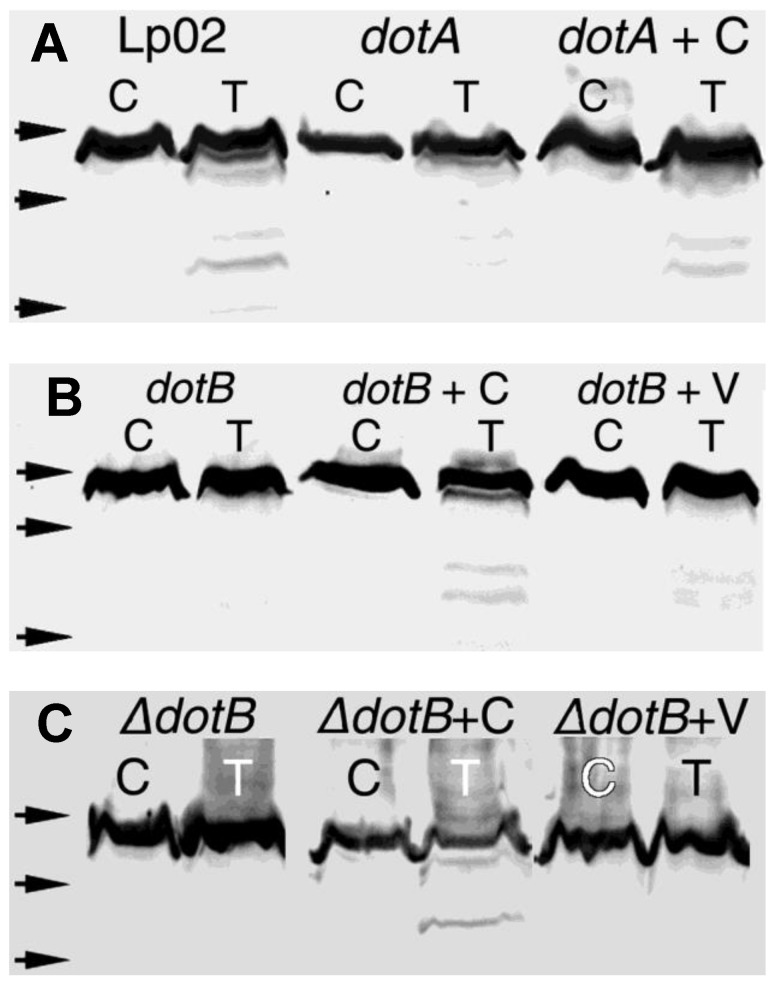
Immunoblots of SDS-PAGE-resolved proteins from whole bacterial cells treated with trypsin (250 µg trypsin/10^9^ bacteria), and immunostained with an HtpB-specific monoclonal antibody and alkaline phosphatase-conjugated rabbit anti-mouse IgG. (**A**) Band patterns produced in the wild-type strain Lp02 and its *dotA*^−^ derivative JV309. Trypsin-accessible HtpB is evidenced by the immunolabeled degradation products of <60 kDa. The presence of trypsin-accessible HtpB was recovered in the *trans*-complemented *dotA*^−^ mutant (*dotA*^−^ + C). (**B**,**C**) Band patterns showing the lack of trypsin-accessible HtpB in two different *dotB* mutants: *dotB*^−^ (with a loss-of-function point mutation), and Δ*dotB* (with a deletion of the *dotB* gene). A recovery of trypsin-accessible HtpB was observed in *trans*-complemented *dotB* mutants (*dotB* + C and Δ*dotB* + C), but not in the mock-complemented mutants carrying an empty vector (*dotB* + V and Δ*dotB* + V). Abbreviations for all panels: “C” = control, trypsin-free sample, “T” = trypsin-treated sample. Arrows at the far left indicate the position of three of the broad-range pre-stained protein size markers (New England Biolabs, Whitby, ON, Canada) corresponding to (from top to bottom) 62, 47 and 37.5 kDa.

**Figure 3 biomolecules-15-00091-f003:**
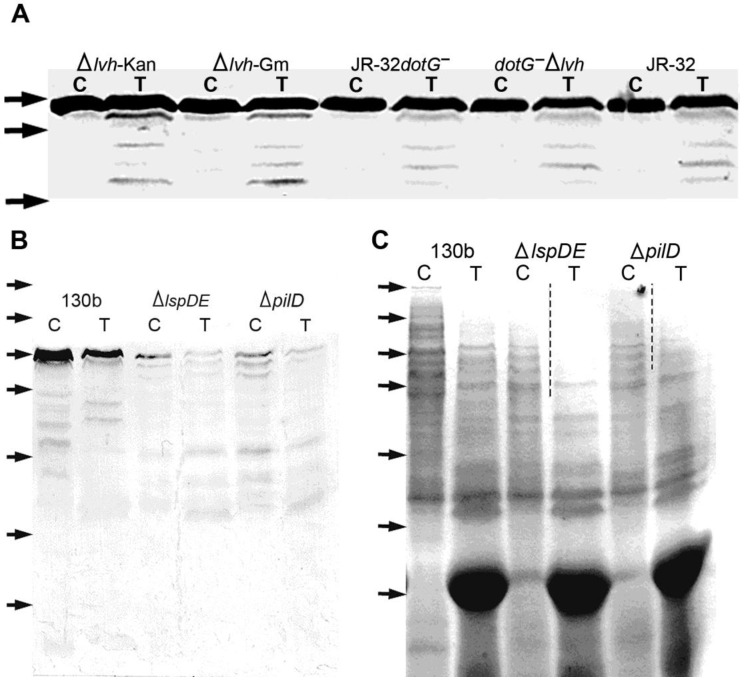
Immunoblots (panels **A**,**B**) and Coomassie Blue-stained SDS-PAGE gel (panel **C**), showing the results of trypsin assays with the indicated *L. pneumophila* strains. Lanes marked “C” contain proteins from control samples of trypsin-free whole bacterial cells, and lanes marked “T” contain proteins from samples of trypsin-treated whole bacterial cells. (**A**) Results of a trypsin accessibility assay with parent strain JR-32 and derivatives. (**B**) Results of a trypsin accessibility assay with parent strain 130b and derivatives. (**C**) Sister gel of the one used for immunostaining in panel B, showing a lack of high molecular weight proteins in the trypsin-treated samples of the Δ*lspDE* and Δ*pilD* mutants (top areas of the lanes marked with the dotted lines). The positions of protein size markers are indicated by arrowheads on the left-side border of each panel. Size markers in (**A**) correspond to 62, 47, and 37.5 kDa. Size markers in (**B**,**C**) correspond to 175, 83, 62, 47, 37.5, 25, and 16.5 kDa. The large Coomassie-stained blobs of protein in the “T” lanes of panel C, correspond to the added trypsin and trypsin inhibitor (both at around 22 kDa).

**Figure 4 biomolecules-15-00091-f004:**
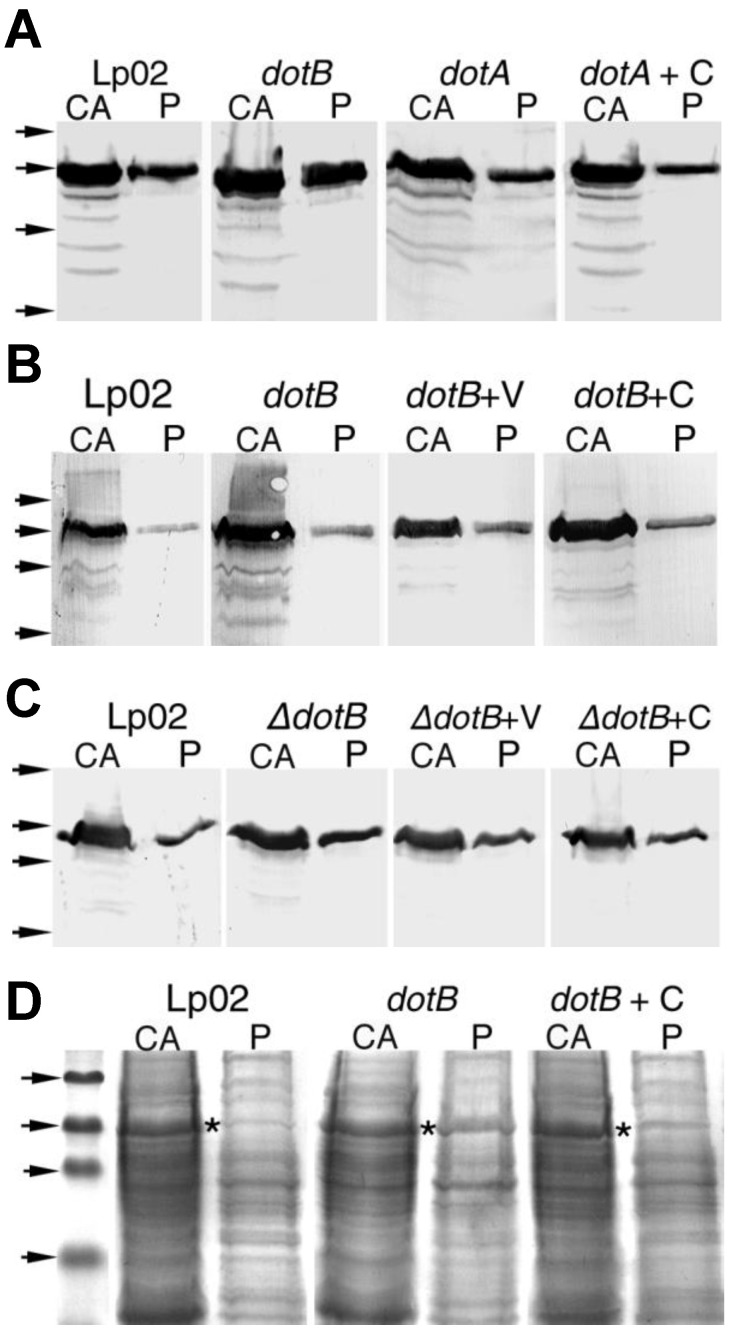
Immunoblots (panels **A**–**C**) and Coomassie-stained SDS-PAGE gel (panel **D**) of *L. pneumophila* samples fractionated by osmotic shock. Each panel represents a separate fractionation experiment. Immunoblots in panels (**A**–**C**) were used to calculate the % periplasmic HtpB shown in Table 8. (**A**) Experiment 1 included the parent strain Lp02 and the *dotA*^−^ and *dotB*^−^ mutants, showing that *dot* mutants have more periplasmic HtpB than Lp02, and that the *trans*-complemented *dotA*^−^ mutant (*dotA*^−^ + C) displays a reduced amount of periplasmic HtpB in relation to the *dotA*^−^ mutant. (**B**) Experiment 2 with parent strain Lp02 and the *dotB*^−^ series. (**C**) Experiment 3 with parent strain Lp02, and the Δ*dotB* series. (**D**) Sister gel run with samples from experiment 2 (panel **B**), and stained with Coomassie Blue to show the overall differences in protein band patterns between lysates (cell associated) and concentrated shockates (periplasmic), as well as to appreciate the ~60 kDa band where HtpB runs (marked with asterisks) in the context of all the other protein bands shown. Abbreviations: CA = Cell-associated; P = Periplasmic. Arrows at the far left of all panels point at the position of protein size markers (from top to bottom): 83, 62, 47, and 37.5 kDa.

**Figure 5 biomolecules-15-00091-f005:**
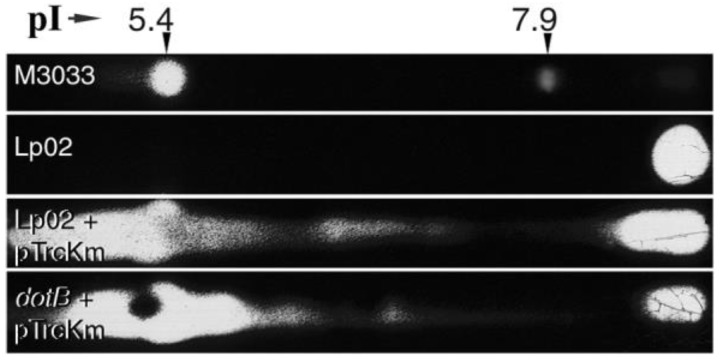
Analysis of β-lactamases in osmotic shockates. Ampicillin-embedded agarose gel showing negatively stained lactamases separated according to their isoelectric point (pI). A whole cell lysate of *E. coli* strain M3033 containing the TEM-1 (pI 5.4) and CTX-M-2 (pI 7.9) β-lactamases was used to provide the reference pIs marked by the arrowheads on the top. The shockate of parent strain Lp02 showed abundant oxacillinase activity at the expected pI > 8.0. Shockates of parent strain Lp02 and of the *dotB* mutant carrying the plasmid pTrcKm show massive TEM-1 activity at pI = 5.4, in addition to the *L. pneumophila* oxacillinase activity at pI > 8.0.

**Figure 6 biomolecules-15-00091-f006:**
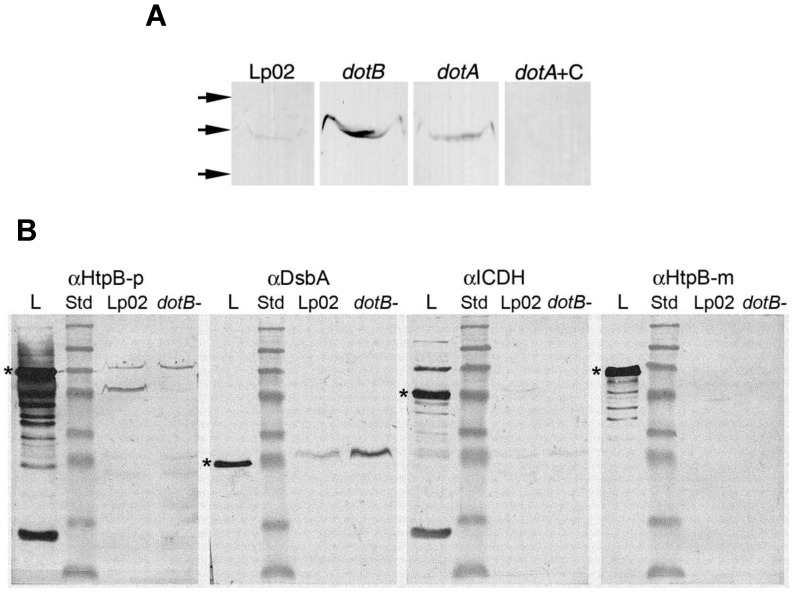
Support for the periplasmic location of HtpB. (**A**) Immunoblot analysis of an SDS-PAGE gel of outer membrane vesicles (OMVs) isolated from the strains shown, showing the increased presence of HtpB in OMVs from the *dot* mutants. Arrows point at the position of protein size markers of (from top to bottom) 83, 62, and 47 kDa. (**B**) Immunostained membranes from SDS-PAGE gels of the samples shown: L = Whole cell lysate of parent strain Lp02; Lp02 = OMVs purified from strain Lp02; and *dotB*^−^ = OMVs purified from the *dotB*^−^ mutant. Each of the four membranes was immunostained with a different antibody as indicated at the top of the panel. The bands corresponding to the full-size target proteins are marked by an asterisk. It should be noted that each gel/membrane was run with the same volume of the same sample, equivalent to 20 μg of total protein. The lanes labeled “Std” contain pre-stained size markers from New England Biolabs (Cat. # P77085), corresponding to (from top to bottom) 175, 80, 58, 46, 30, 25, 17 and 7 kDa.

**Figure 7 biomolecules-15-00091-f007:**
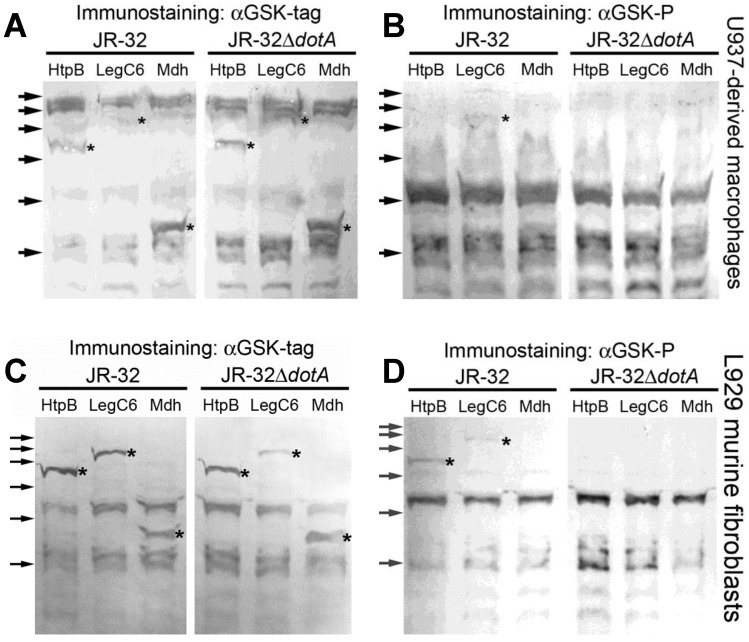
Translocation of HtpB to mammalian host cells. (**A**,**B**) Immunoblots of cell lysates of U937-derived human macrophages infected with JR-32 or JR-32Δ*dotA* derivatives expressing the GSK-tagged proteins HtpB (test protein), LegC6 (positive translocation control), and Mdh (negative translocation control). Immunoblots were either stained with the αGSK-tag antibody (which recognizes the non-phosphorylated GSK tag) (panel **A**), or with the αGSK-P antibody (which recognizes the phosphorylated GSK tag) (panel **B**). (**C**,**D**) Immunoblots (as described for panels (**A**,**B**)) but with cell lysates of L929 mouse fibroblasts. The target immunostained bands of interest are marked with an asterisk. The arrowheads on the left border of each immunoblot mark the position of pre-stained New England Biolabs protein size markers (P7719S) corresponding to (from top to bottom): 130, 95, 72, 55, 43, and 34 kDa.

**Figure 8 biomolecules-15-00091-f008:**
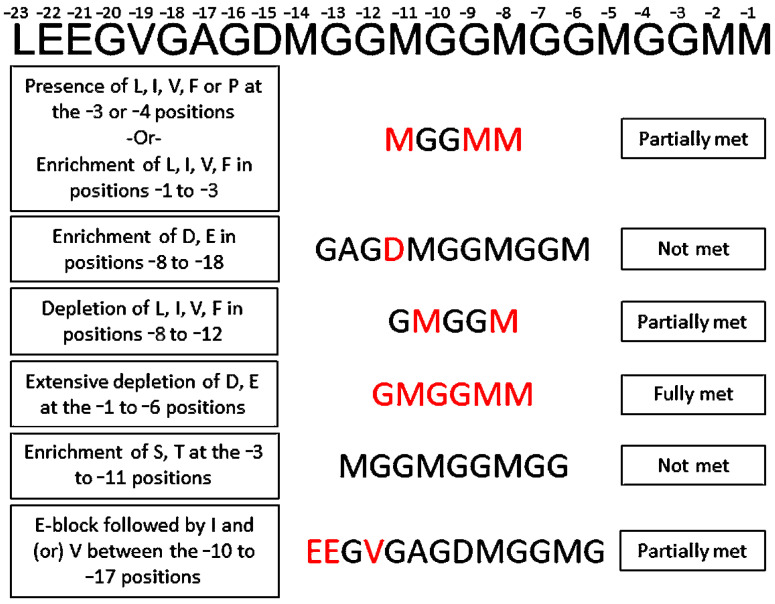
Diagrammatic representation of the motifs characteristic of Dot/Icm effectors, as they apply to the C-term of HtpB. The amino acid sequence of HtpB’s C-term is shown at the top of the figure in single letter format, with the position of each residue marked on top. The six major motifs that have been used as predictors for secretion by the Dot/Icm system are listed in the column of boxes on the left side of the figure. In the central column, the partial HtpB C-term amino acid sequences that are relevant for the corresponding predictor on the left are shown. The amino acid residues marked in red represent those that meet the predictor. The column of boxes on the right side of the figure simply state whether the corresponding predictor was fully met, not met, or partially met.

**Figure 9 biomolecules-15-00091-f009:**
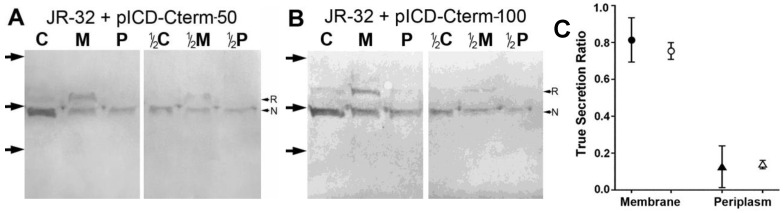
The HtpB C-term mediates membrane association. (**A**) Immunoblots stained with antibody αICDH of fractionated JR-32 cells carrying plasmid pICD-Cterm-50, which express the recombinant ICDH form with the added 50 last amino acids of the HtpB C-term. The fractions are labeled as “C” for cytoplasm, “M” for membrane, and “P” for periplasm. The amount of protein per lane (20 μg) was halved in the immunoblot at the right side of panel (**A**), to ensure that the densitometry band values used for the calculation of the True Secretion Ratios shown in panel C were within a linear range. The arrowheads shown at the left border of the panel represent protein size markers of (top to bottom) 72, 55, and 43 kDa. The small arrowheads on the right border of the panel point at the position of the recombinant (R) or native (N) forms of ICDH. (**B**) Same as described for panel A, but with immunoblots of fractionated JR-32 cells carrying plasmid pICD-Cterm-100, which express the recombinant ICDH form with the added 100 last amino acids of the HtpB C-term. (**C**) Graph showing the True Secretion Ratios of recombinant ICDH forms (in relation to native ICDH) in the membrane (circles) and periplasm (triangles) fractions. The solid black symbols are for the recombinant ICDH-HtpB-Cterm 50, and the open symbols are for the recombinant ICDH-HtpB-Cterm 100. Graph points represent mean values ± std. deviation (vertical bars) of three independent experiments (n = 3).

**Figure 10 biomolecules-15-00091-f010:**
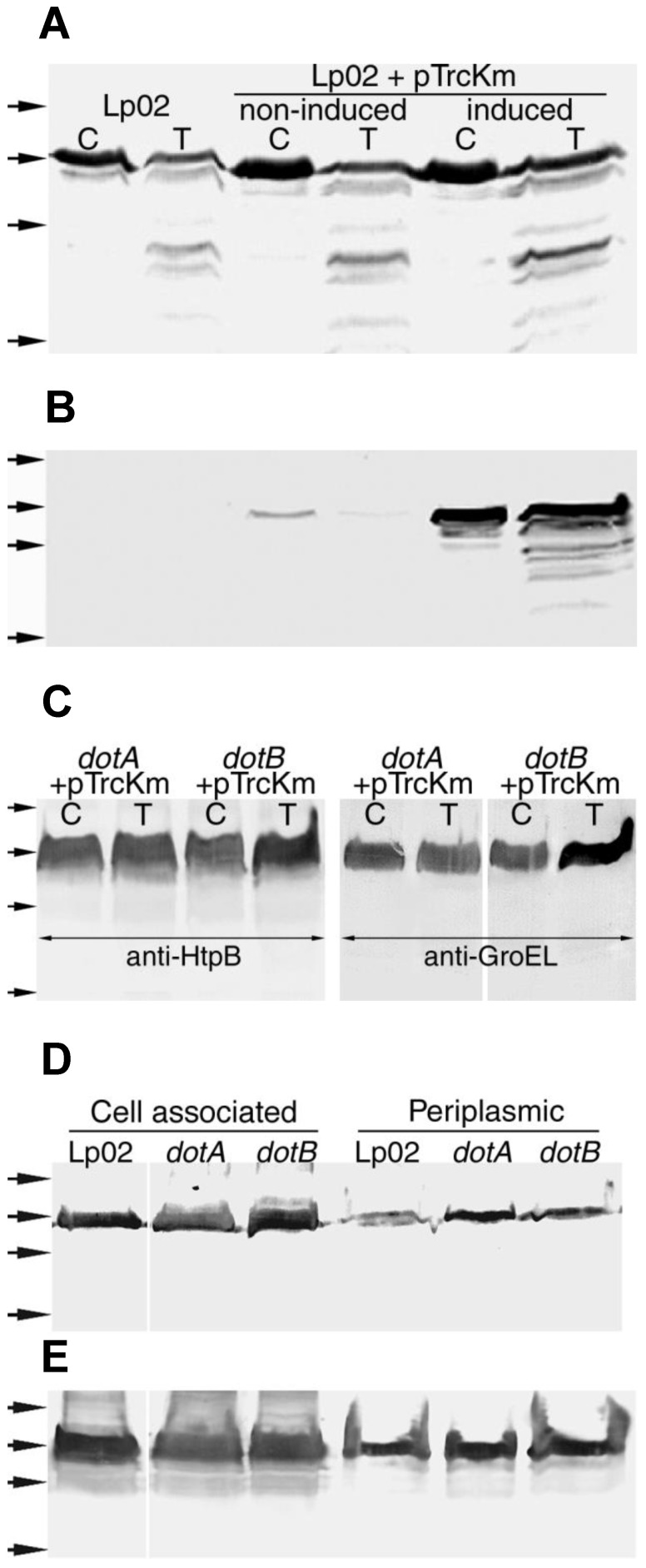
Immunoblot analysis of HtpB and GroEL in *L. pneumophila* variants expressing GroEL. (**A**) Results from a trypsin assay performed with Lp02 and Lp02 carrying pTrcKm (from which GroEL is expressed upon induction with IPTG). Blot was immunostained with αHtpB-m. (**B**) Twin immunoblot of (**A**) stained with αGroEL monoclonal antibody. (**C**) Results from a trypsin assay performed with *dotA*^−^ and *dotB*^−^ mutants carrying pTrcKm (expressing GroEL). (**D**) Immunoblot analysis of cell-associated HtpB (from cell lysates) and periplasmic HtpB (from concentrated shockates) in Lp02, and the *dotA*^−^ and *dotB*^−^ mutants expressing GroEL from plasmid pTrcKm. (**E**) Twin immunoblot of (**D**) immunostained with αGroEL. Samples in panels (**D**,**E**) correspond to data shown in Table 10. In all panels, arrows point at the position of protein size markers of (from top to bottom) 83, 62, 47, and 37.5 kDa. Abbreviations: C = control trypsin-free samples; T = trypsin-treated samples.

**Figure 11 biomolecules-15-00091-f011:**
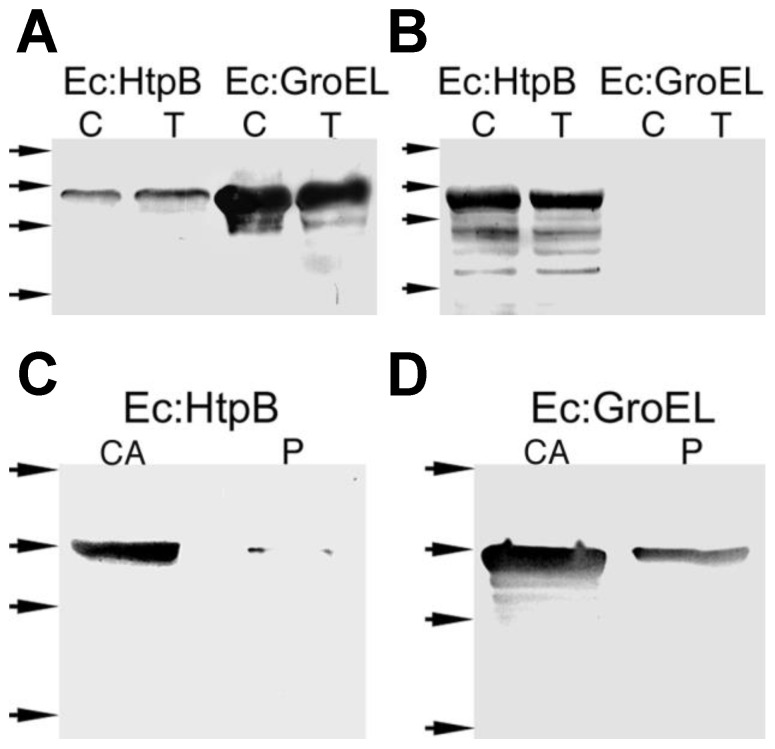
Immunoblot analysis of trypsin accessible (twin panels **A**,**B**) or periplasmic Hsp60 (twin panels **C**,**D**) in *E. coli* JM109 expressing HtpB from plasmid pSH16 (designated Ec:HtpB), or overexpressing GroEL from plasmid pTrcGroE (designated Ec:GroEL). Panels (**A**,**D**) were immunostained with monoclonal antibody αGroEL, and panels (**B**,**C**) with αHtpB-m. C = control, trypsin-free samples; T = trypsin-treated samples; CA = cell-associated Hsp60 from lysates; P = periplasmic Hsp60 from shockates. Arrows point at the position of protein size markers of (from top to bottom) 83, 62, 47, and 37.5 kDa.

**Table 1 biomolecules-15-00091-t001:** List of bacterial strains used. *L. pneumophila* parental strains (**in boldface**) and derivatives are listed first, followed by *Escherichia coli* strains.

Name	Designation	Description	Reference/Source
**Lp02**	**Lp02**	Virulent, salt-sensitive, restriction minus, thymidine auxothroph, streptomycin^R^, serogroup 1, Philadelphia-1 strain.	[14]/Dr. R.R. Isberg ^1^
JV303	*dotB* ^−^	Lp02 derivative, *dotB* mutant carrying the mutation E312V, salt-tolerant, avirulent.	Dr. J.P. Vogel ^2^
JV1119	*dotB*^−^ + V	JV303 carrying plasmid pJB908 (see Table 2), which is the *trans*-complementation empty vector.	Dr. J.P. Vogel
JV1163	*dotB*^−^ + C	JV303 carrying plasmid pJB1153 (see Table 2),which is the *dotB trans*-complementing plasmid.	Dr. J.P. Vogel
JV918	Δ*dotB*	Lp02 derivative lacking the *dotB* open reading frame (*dotB* deletion mutant), salt-tolerant, avirulent.	Dr. J.P. Vogel
JV1133	Δ*dotB* + V	JV918 carrying plasmid pJB908 (see Table 2), which is the *trans*-complementation empty vector.	Dr. J.P. Vogel
JV1170	Δ*dotB* + C	JV918 carrying plasmid pJB1153 (see Table 2),which is the *dotB trans*-complementing plasmid.	Dr. J.P. Vogel
JV309	*dotA* ^−^	Lp02 derivative, *dotA* mutant, salt tolerant, avirulent.	[14]/Dr. R.R. Isberg
JV309 compl.	*dotA*^−^ + C	JV309 carrying plasmid pKB9 (see Table 2), which is the *dotA trans*-complementing plasmid.	[31]/Garduño Lab
**JR-32**	**JR-32**	Virulent, salt-sensitive, streptomycin^R^, restriction-deficient, modification-positive derivative of strain AM511, serogroup 1, Philadelphia 1 strain.	[13]/Dr. H.A. Shuman ^3^
JR-32Δ*dotA*	Δ*dotA*	JR-32 derivative carrying a Kan^R^ cassette insertion replacing the middle of the *dotA* open reading frame.	This study
GS-28K	Δ*lvh*-Kan	JR-32 derivative carrying a Kan^R^ cassette insertion replacing the entire *lvh* region.	[32]/Dr. H.A. Shuman
GS-28G	Δ*lvh*-Gm	JR-32 derivative carrying a Gm^R^ cassette insertion replacing the entire *lvh* region.	[32]/Dr. H.A. Shuman
LELA4432	JR-32*dotG*^−^	JR-32 derivative with the insertion *icmE4432*::Tn*903*dII*lacZ* (*dotG/icmE* mutant).	[16]/Dr. H.A. Shuman
LELA4432-28	*dotG*^−^Δ*lvh*	LELA4432 derivative carrying a Gm^R^ cassette insertion replacing the entire *lvh* region.	[32]/Dr. H.A. Shuman
**130b**	**130b**	Virulent, serogroup 1 clinical isolate from the Wadsworth Veterans Administration Hospital, Los Angeles CA, USA.	[33]/Dr. N.P. Cianciotto ^4^
NU243	Δ*pilD*	130b derivative carrying a Kan^R^ cassette insertion in the prepilin peptidase gene *pilD*.	[34]/Dr. N.P. Cianciotto
NU258	Δ*lspDE*	130b derivative carrying a Kan^R^ cassette insertion in the *lspDE* operon.	[35]/Dr. N.P. Cianciotto
*E. coli* DH5α	DH5α	Lab strain used for cloning: F^−^ Φ80 Δ*lacZ* ΔM15 Δ(*lacZYA*-*argF*) *U169 supE44 hsdR17 recA1 endA1 gyrA96 thi*-*1 relA1 deoR*	Clontech, Mountain View, CA, USA
*E. coli* BL-21	BL-21	Strain used for protein expression: *fhuA2 [lon] ompT gal [dcm]* Δ*hsdS*, protease deficient, lacks the T7 RNA polymerase. Commercially available from New England Biolabs (NEB)	NEB, Ipswich MA, USA/Dr. John Rohde ^5^
*E. coli* Δ*phoA*	Δ*phoA*	Derivative of BL-21 with an in-frame marker-less deletion of *phoA*	Dr. John Rohde
*E. coli* JF626	JF626	F’ (*traD36 proAB^+^ lacI*^q^ *lacZ* Δ*M15)* Δ(*lac pro*) *thi rpsL supE endA sbcB15 hsdR4*	[36]/Dr. P.S. Hoffman ^6^
*E. coli* JM109	JM109	K-12 derivative, Rec^−^ F’ [*traD*36 *proAB*^+^ *lacI*^q^ *lacZ*Δ*M15*] *endA1 recA1 hsdR17 supE44 thi-1 gyrA96* Δ(*lac-proAB*), shows the r_K_^−^ phenotype	[36]/Dr. P.S. Hoffman
*E. coli* M3033	M3033	Used as reference for lactamases CTX-M-2 and TEM-1	[37]/Dr. R. Melano ^7^

^1^ Dr. Ralph R. Isberg, Department of Molecular Biology and Microbiology, TUFTS University Medical School, Boston MA, USA. ^2^ Dr. Joseph P. Vogel, Department of Molecular Microbiology, Washington University School of Medicine, St. Louis, Missouri, USA. ^3^ Dr. Howard A. Shuman, Department of Microbiology, College of Physicians & Surgeons, Columbia University, New York NY, USA. ^4^ Dr. Nicholas P. Cianciotto, Department of Microbiology and Immunology, Northwestern University, Chicago, Illinois, USA. ^5^ Dr. John Rohde, Department of Microbiology and Immunology, Dalhousie University, Halifax NS, Canada. ^6^ Dr. Paul S. Hoffman, Department of Microbiology and Immunology, Dalhousie University. Currently at the University of Virginia, Charlottesville, VA, USA. ^7^ Dr. Roberto Melano, Dept. of Microbiology and Immunology, Dalhousie University. Currently at the Public Health Ontario Laboratories, Toronto ON, Canada.

**Table 2 biomolecules-15-00091-t002:** List of plasmids used or created (in alphabetical order).

Name	Description	Reference/Source
pBluescript II KS or SK	High copy number plasmid used as a general cloning vector in *E*. *coli*, Amp^r^.	Stratagene, La Jolla CA, USA
pBRDX	Suicide delivery vector carrying Cm^R^ and two counter-selectable markers: *sacB* (confers sensitivity to sucrose) and *rdxA* (confers sensitivity to metronidazole).	[38]/Dr. P.S. Hoffman ^1^
pICD-Cterm-50	Derivative of pMMB carrying a fusion of the *icdH* gene (encoding the enzyme iso-citrate dehydrogenase) with the sequence encoding the last 50 amino acids of HtpB.	This study
pICD-Cterm-100	Derivative of pMMB carrying a fusion of the *icdH* gene (encoding the enzyme iso-citrate dehydrogenase) with the sequence encoding the last 100 amino acids of HtpB	This study
pIP100	Derivative of pMMB carrying a synthetic *phoA* gene lacking its N-term secretion signal and with a C-term in-frame fusion with the last 100 codons of *htpB*.	This study
pJB908	Derivative of pKB5 in which the RSF1010 origin of transfer (base pairs 2623 to 2748) of pKB5 has been deleted.	Dr. J.P. Vogel ^2^
pJB1153	Derivative of pJB908 carrying a *Bam*H1-*Sal*1 fragment containing the complete *dotB* open reading frame from Lp02.	Dr. J.P. Vogel
pKB5	Derivative of pTZDi and pMMB67EH carrying the thymidylate synthetase gene *tdi* from bacteriophage T4.	[14]/Dr. R.R. Isberg ^3^
pKB9	Derivative of pKB5 containing the complete *dotA* open reading frame from Lp02 under the control of the inducible *tac* promoter	[19]/Dr. R.R. Isberg
pMGHtp	Derivative of pMMB carrying a 5′ fusion of the *htpB* open reading frame (*lpg0688*) with the sequence encoding 13 amino-acids of the eukaryotic reporter protein GSK3β	This study
pMGLeg	Derivative of pMMB carrying a 5′ fusion of the *legC6* open reading frame (*lpg1588*) with the sequence encoding 13 amino-acids of the eukaryotic reporter protein GSK3β	This study
pMGMdh	Derivative of pMMB carrying a 5′ fusion of the *mdh* open reading frame (*lpg2352*) with the sequence encoding 13 amino-acids of the eukaryotic reporter protein GSK3β	This study
pMHtp6His	Derivative of pMMB carrying a 3′ in-frame fusion of the complete *htpB* gene (*lpg0688*) with a sequence encoding 6 histidine residues (added to the C-term of HtpB)	This study
pMMB	Original designation pMMB207c. Derivative of RSF1010 (IncQ *oriT lacI*^q^ P*_tac_* [IPTG-inducible promoter]) with a deletion of *mobA* (Δ*mobA*), Amp^r^, Cm^r^.	[39,40]/Dr. H.A. Shuman ^4^
pSH16	Derivative of pUC19 carrying a 3.2 kilobase *Eco*RI fragment containing the complete *htpAB* operon from *L. pneumophila* strain Lp-Svir.	[41]/Dr. P.S. Hoffman
pTrcGroE	Derivative of pTrc99a (Amersham Pharmacia Biotech) carrying the *E. coli groELS* operon under the control of the inducible *trc* promoter. Has the Amp^R^ marker.	Dr. P.B. Sigler ^5^
pTrcKm	Derivative of pTrcGroE carrying the Km-resistance cassette from p34S-Km3 inserted into the *Hind*III site of its MCS.	This study
p34S-Km3	Carries a kanamycin-resistance cassette engineered to lack internal restriction sites and flanked by numerous restriction sites.	[42]/Dr. J.J. Dennis ^6^

^1^ Dr. Paul S. Hoffman, Department of Microbiology and Immunology, Dalhousie University. Currently at the University of Virginia, Charlottesville, VA, USA. ^2^ Dr. Joseph P. Vogel is at the Department of Molecular Microbiology, Washington University School of Medicine, St. Louis, Missouri, USA. ^3^ Dr. Ralph R. Isberg is a Howard Hughes Medical Institute investigator at the Department of Molecular Biology and Microbiology, TUFTS University Medical School, Boston, MA. ^4^ Dr. Howard A. Shuman is at the Department of Microbiology, College of Physicians & Surgeons, Columbia University, New York, NY. ^5^ Dr. Paul B. Sigler is a Howard Hughes Medical Institute investigator at the Molecular Biophysics & Biochemistry Department, Yale University, New Haven, CT. ^6^ Dr. Jonathan J. Dennis is at the Department of Biological Sciences, University of Alberta, Edmonton, AB.

**Table 3 biomolecules-15-00091-t003:** Antibodies used (listed in alphabetical order) for Western blotting (WB) and (or) immunogold labeling (IGL).

Name	Description and Use	Reference/Source
αDsbA2	Mouse polyclonal hyperimmune serum raised vs. the DsbA2 protein from *L. pneumophila*. Diluted 1:5000 for WB.	[52]/Dr. P.S. Hoffman ^1^
αGroEL	Commercial mouse monoclonal. Stressgen SPA-870. Diluted 1:1000 for WB.	SressGen Assay Designs ^2^
αGSK-P	Commercial rabbit monoclonal., #9336. Diluted 1:1000 for WB. Recognizes the phosphorylated GSK tag.	Cell Signaling Technology ^3^
αGSK-tag	Commercial rabbit monoclonal., #9315. Diluted 1:1000 for WB. Recognizes the native GSK tag.	Cell Signaling Technology
αHtpB-p	Rabbit polyclonal hyperimmune serum raised either vs. HtpB from *L. pneumophila* or vs. recombinant HtpB from *E. coli*. Diluted 1:500 for IGL or WB.	[26,41]/Garduño Lab
αHtpB-m	Mouse monoclonal. Original designation MAbGW2X4B8B2H6. Used as hybridoma cell culture supernatant, undiluted for IGL, or diluted 1:100 for WB.	[53]/Dr. P.S. Hoffman
αICDH	Rabbit polyclonal hyperimmune serum raised vs. the isocitrate dehydrogenase from *B. subtilis* Diluted 1:5000 for WB.	[54]/Dr. A.L. Sonenshein ^4^
αMouseAP	Commercial rabbit purified immunoglobulin G conjugated to alkaline phosphatase. Diluted 1:5000 for WB	Cedarlane Labs ^5^
αMouseGold	Rabbit purified immunoglobulin G conjugated to 10 nm gold particles. Obtained commercially. Diluted 1:100 for IGL	Sigma Immunochemicals ^6^
αRabbitAP	Goat purified immunoglobulin G conjugated to alkaline phosphatase. Obtained commercially. Diluted 1:5000 for WB	Cedarlane Labs
αRabbitGold	Goat purified immunoglobulin G conjugated to 10 nm gold particles. Obtained commercially. Diluted 1:100 for IGL	Sigma Immunochemicals
α6XHis-tag	Commercial mouse monoclonal Ab18184. Diluted 1:1000 for IGL or WB	Abcam ^7^

^1^ Dr. Paul S. Hoffman, Department of Microbiology and Immunology, Dalhousie University. Currently at the University of Virginia, Charlottesville VA, USA. ^2^ The company originally named Stressgen, changed to StressGen-Assay Designs, which, at the time we purchased the antibody, was located in Ann Arbor MI, USA. An equivalent antibody (with a different designation) can be currently obtained from MyBioSource (San Diego CA, USA). ^3^ Cell Signaling Technology, Danvers MA, USA. ^4^ Dr. Abraham L. Sonenshein, Department of Molecular Biology and Microbiology, Tufts University School of Medicine, Boston MA, USA. ^5^ Cedarlane Laboratories, located in Burlington ON, Canada. ^6^ Sigma Immunochemicals is a Division of Sigma-Aldrich Canada Co., Oakville ON, Canada. ^7^ Abcam corporate headquarters are located in Cambridge, UK.

**Table 4 biomolecules-15-00091-t004:** Dimensions of the typical 2D sections ^1^ of various *L. pneumophila* strains.

Strains	Area of Cytoplasm (in μm^2^)	Length of CM (in μm)	Area of Periplasm (in μm^2^)	Length of OM (in μm)
Lp02	0.12 ± 0.09	1.46 ± 0.85	0.10 ± 0.07	1.85 ± 0.92
JR-32	0.15 ± 0.04	1.70 ± 0.66	0.14 ± 0.02	2.40 ± 0.32
Lp1-SVir	0.15 ± 0.02	1.56 ± 0.13	0.08 ± 0.01	1.91 ± 0.14
2064	0.25 ± 0.09	1.88 ± 0.40	0.10 ± 0.02	2.20 ± 0.41

^1^ The typical sections for Lp02 and derivatives were calculated from measurements performed in 210 bacterial cell sections from 7 labeling experiments. The typical sections for JR-32 and derivatives were calculated from measurements performed in 120 bacterial cell sections (from 4 labeling experiments), whereas the typical sections for Lp1-SVir and 2064 included 720 and 180 bacterial cell sections, respectively. The dimensions given apply to both the parent strains and their derivative mutants. The data for strains Lp1-SVir and 2064 (which were not analyzed in this study) were taken from Garduño et al. [24] and are included here only as a reference.

**Table 5 biomolecules-15-00091-t005:** Distribution of HtpB epitopes in typical sections of *L. pneumophila* strain Lp02 and derivatives (given in Table 4). One labeling experiment per strain was conducted, and results are shown as a single compounded number of particles per typical cell compartment, as well as percent ratios (in parentheses).

Compartment ^1^	Lp02	*dotA* ^−^	*dotB* ^−^	Δ*dotB*	Δ*dotB* + V	Δ*dotB* + C
Cytoplasm	8.9 (11.5)	12.1 (14.7)	15.5 (7.7)	10.9 (10.8)	11.7 (14.8)	13.5 (21.5)
Cytopl. Membr.	6.6 (8.5)	6.6 (8.0)	12.4 (6.2)	8.8 (8.7)	7.4 (9.4)	5.3 (8.5)
Periplasm	22.1 (28.5)	36.9 (44.9)	74.0 (36.8)	63.2 (62.8)	44.6 (56.4)	26.6 (42.4)
OM-Surface	40.0 (51.5)	26.6 (32.4)	99.2 (49.3)	17.8 (17.7)	15.3 (19.4)	17.3 (27.6)
Total	77.6 (100)	82.2 (100)	201.1 (100)	100.7 (100)	79.0 (100)	62.7 (100)

^1^ Abbreviations: Cytopl. Membr. = cytoplasmic membrane, OM = outer membrane.

**Table 6 biomolecules-15-00091-t006:** Distribution of HtpB epitopes in typical sections of *L. pneumophila* strain JR-32 and derivatives (given in Table 4). One labeling experiment per strain was conducted, and results are shown as a single compounded number of particles per typical cell compartment, as well as percent ratios (in parentheses).

Compartment ^1^	JR-32	Δ*lvh*-Gm	JR-32*dotG*^−^	*dotG*^−^Δ*lvh*
Cytoplasm	4.7 (4.5)	3.9 (6.3)	2.6 (6.2)	10.8 (10.0)
Cytopl. Membr.	5.6 (5.3)	7.4 (12.1)	6.6 (15.9)	16.6 (15.3)
Periplasm	25.5 (24.3)	16.5 (26.9)	11.4 (27.5)	16.8 (15.5)
OM-Surface	69.0 (65.9)	33.6 (54.7)	20.9 (50.4)	64.1 (59.2)
Total	104.8 (100)	61.4 (100)	41.5 (100)	108.3 (100)

^1^ Abbreviations: Cytopl. Membr. = cytoplasmic membrane, OM = outer membrane.

**Table 7 biomolecules-15-00091-t007:** Surface-exposed HtpB epitopes on whole intact bacterial cells labeled in suspension.

Strain	% Labeled Bacteria	Number of Epitopes per 50 Cells	Mean # of Epitopes/Bacterium
Lp02	100	555	11.1
Lp02 *dotA*^−^	82	250	5.0
Lp02 *dotB*^−^	94	260	5.2

**Table 8 biomolecules-15-00091-t008:** Activity of glucose-6-phosphate dehydrogenase (G6PDH), and densitometry data of immuno-stained HtpB bands, in lysates or concentrated shockates used to estimate the percentage of periplasmic HtpB in different *L. pneumophila* Lp02 derivatives. The % ratio of periplasmic G6PDH activity represents the % ratio of periplasm contamination with cytoplasmic content. The % contamination was subtracted from the % ratio of periplasmic HtpB to calculate the % of true periplasmic HtpB.

Strain	Total G6PDH Activity ^1^		Contamination (%)	Total Band Density ^2^		True Periplasmic HtpB (%)
	Cell Associated	Periplasmic		Cell Associated	Periplasmic	
Fractionation experiment corresponding to the immunoblots shown in Figure 4A						
Lp02	111.9	0	0	28,675	630	2.2
*dotB* ^−^	108.2	2.2	2.0	25,900	2450	7.2
*dotA* ^−^	62.0	0	0	25,900	1120	4.3
*dotA*^−^ + C	74.9	2.3	3.0	24,975	893	0.6
Fractionation experiment corresponding to the immunoblots shown in Figure 4B						
Lp02	50.1	0	0	70,070	770	1.1
*dotB* ^−^	41.2	0.3	0.7	72,358	1848	1.9
*dotB*^−^ + V	41.2	0.6	1.4	88,660	2048	0.9
*dotB*^−^ + C	46.1	0.4	0.9	91,520	1201	0
Fractionation experiment corresponding to the immunoblots shown in Figure 4C						
Lp02	170.5	0	0	158,950	2599	1.6
Δ*dotB*	192.5	0	0	178,200	3752	2.1
Δ*dotB* + V	116.6	0	0	138,325	2532	1.8
Δ*dotB* + C	231.3	0	0	151,250	1840	1.2

^1^ Total G6PDH activity was estimated from the specific activity (μmoles of NADP reduced per minute per mg of protein) multiplied by the total mg of protein in lysates (cell associated) or shockates (periplasmic). ^2^ Total band density corresponds to the IOD values of the immuno-stained bands shown in Figure 4 divided by 40 (μg of protein per lane) and multiplied by the total protein (in μg) present in lysates (cell-associated) or in shockates (periplasmic). In turn, total protein = (protein concentration) × (total volume of shockate or lysate).

**Table 9 biomolecules-15-00091-t009:** Percent ratios of true periplasmic HtpB and DsbA2 in different *L. pneumophila* Lp02 derivatives, based on immunoblotting ^1^. The % ratio of periplasmic ICDH is the shockate contamination value, used to calculate the % ratios of true periplasmic (TP) HtpB or DsbA2. OD = optical density, Exp = experiment, ND = Not determined.

	ICDH Total OD		% Ratio	HtpB Total OD		% Ratio	DsbA2 Total OD		% Ratio
	Shockate	Lysate	Contam.	Shockate	Lysate	TP HtpB	Shockate	Lysate	TP DsbA2
Lp02 Exp1	891	103,125	0.86	302	50,000	0	ND	ND	ND
*dotB*^−^ Exp1	994	334,875	0.30	365	99,875	0.07	ND	ND	ND
Lp02 Exp2	44.9	80,192	0.06	266	151,818	0.11	449	85,920	0.46
*dotB*^−^ Exp2	125.6	90,825	0.14	1166	308,779	0.24	1110	151,375	0.59
*dotB*^−^ Exp3	640	386,000	0.17	1862	878,150	0.04	2968	559,700	0.36
*dotB*^−^ + V	443	189,376	0.23	1519	307,736	0.26	1139	165,704	0.46
*dotB*^−^ + C	823	146,190	0.56	2288	425,280	0	1739	139,545	0.69

^1^ Densitometry data, as well as protein concentrations and total volumes of shockates and lysates, are presented in Appendix A.

**Table 10 biomolecules-15-00091-t010:** Activity of glucose-6-phosphate dehydrogenase (G6PDH), and densitometry data of immuno-stained Hsp60 bands, in lysates or concentrated shockates used to estimate the percentage of periplasmic Hsp60 in different *L. pneumophila* Lp02 derivatives expressing recombinant GroEL, or *E. coli* strain JM109 expressing recombinant HtpB or overexpressing GroEL. The % ratio of periplasmic G6PDH activity represents the % ratio of periplasm contamination with cytoplasmic content. The % contamination was subtracted from the % ratio of periplasmic Hsp60, to calculate the % of true periplasmic Hsp60. In the “Total band density” and the “True periplasmic Hsp60” columns, the values are split for GroEL (and HtpB in parenthesis).

Strain	Total G6PDH Activity ^1^		Contamination (%)	Total Band Density ^2^ × (10^3^)		True Periplasmic Hsp60 (%)
	Cell Associated	Periplasmic		Cell Associated	Periplasmic	
Fractionation experiment corresponding to the immunoblots shown in Figure 10D,E						
Lp02 + pTrcKm	81.8	0.29	0.4	465 (27.2)	14.0 (0.4)	2.6 (1.1)
*dotA*^−^ + pTrcKm	65.8	0.16	0.2	317 (27.2)	13.1 (0.9)	3.9 (3.1)
*dotB*^−^ + pTrcKm	87.0	0.29	0.3	335 (34.1)	18.1 (0.6)	5.1 (1.5)
Fractionation experiment corresponding to the immunoblots shown in Figure 11C,D						
JM109 + pSH16	700	0.30	0.04	(244)	(0.08)	(0)
JM109 + pTrcKm	2473	1.50	0.06	9569	54.8	0.51

^1^ Total G6PDH activity was estimated from the specific activity (μmoles of NADP reduced per minute per mg of protein) multiplied by the total mg of protein in lysates (cell associated) or shockates (periplasmic). ^2^ Total band density corresponds to the IOD values of the immunostained bands shown in Figure 10 or Figure 11, divided by 40 (μg of protein per lane) and multiplied by the total protein (in μg) present in lysates (cell associated) or in shockates (periplasmic). In turn, total protein = (protein concentration) × (total volume of shockate or lysate).

## Data Availability

Materials and data are available through direct contact with P.R. and (or) R.A.G.

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
