# Peer review of "The Passage of Chaperonins to Extracellular Locations in Legionella pneumophila Requires a Functional Dot/Icm System"

_biomolecules, 2025, doi:10.3390/biom15010091_

Round 1

Reviewer 1 Report

Comments and Suggestions for Authors

In this study, the authors report that the chaperonin HtbB of Legionella pneumophila is secreted extracytoplasmically via the novel type four secretion system (Dot/Icm). HtpB is the Lpn equivalent of GroEL the common heat shock protein Hsp60 of E. coli.  In E. coli, both GroEL and HtpB reside in the cytoplasm, while in Legionella, they are both found extracytoplasmically suggesting a novel secretion mechanism present in Lpn and absent in E. coli.  Building on considerable prior work, Dr. Garduno’s group focus now on how HtbB traverses the cytoplasmic membrane and then how it passes across the outer membrane and eventually into the cytoplasmic vacuole (LCV) of host cells.   The results of these studies are highly dependent on interpretation of cellular localization data from both immunogold studies of ultrathin sections and on enzyme markers of known cellular location.  Finally, the studies employ a range of Dot/Icm mutants (dotA, dotB and IcmE/dotG and their complements).  The authors have gone to great lengths to provide a foundation for measures and counting measures to dissect out cellular location and relative differences between wild type and mutants. It is also fascinating to think that the Dot/Icm secretion system has evolved to recognize highly conserved amino acid sequences in HtpB and GroEL to enable secretion of a protein that is not predicted to reside outside the cytoplasm. The studies are carefully done and the methods used are sufficiently detailed to instill confidence and the authors are careful in interpretation of findings.  Overall, the findings are significant and provide a solid foundation for future research to delineate at a more molecular level.  The manuscript is lengthy, but well written.  Below, I list a few concerns for attention by the authors. 

Comments.

The authors might look at some of the methods, such as the Bradford protein determination, and just cite rather than provide details. 

The semiquantitative methods with immunogold, tables of data etc, an absence of mean and standard deviation should be explained so readers understand that 30 or more cells are evaluated for each data point presented as a mean.  Maybe explain this in the methods section so that readers don’t stumble through this while reading through table legends.

Table 3 and elsewhere, with respect to antibody against DsbA2 –to avoid confusion be sure that it is written as DsbA2 and not DsbA as these are distinct proteins.

One of the perversities of working with Dot/Icm mutants is that the proteins interact with other members of the Dot/Icm secretion system and with many effector proteins. Ordinarily, the type of mutations used might not matter, but since some mutants might produce truncated proteins that still interact with other proteins, interpretation of results with these mutants might not be straightforward.  Some of this is seen between dotB and delta dotB mutants.  The absence of the protein might be important and the authors comment on this to some extent in the discussion section.  Also, in the case of IcmE/DotG, it is a very large protein and so the use of mutants containing complete deletions of the gene might provide cleaner results.  This is also discussed in the discussion section on DotG. This reviewer worries that it may not be known where the Tn903dIIlacZ insertion is in strain LELA4432.  In this respect, the authors may not be aware of Kpadeh et al (Mol Microbiol. 2015 Mar;95(6):1054-69) where it was found that a complete deletion of dotG had no effect on virulence and rendered the Lp02 strain and JV3559 hyper hemolytic and hyper virulent while strain JV573 expressing a truncated DotG containing an internal deletion was avirulent.  The hyper hemolytic phenotype might suggest HtpB is more efficiently gaining surface location – as mentioned in citation 99.  If it is still true that DotG spans the periplasm or forms a collar as most recently suggested, a truncation might plug the secretion system yielding the avirulent phenotype and a different interpretation with respect to a complete deletion. 

Author Response

Reviewer #1, Comment 1

In this study, the authors report that the chaperonin HtbB of Legionella pneumophila is secreted extracytoplasmically via the novel type four secretion system (Dot/Icm). HtpB is the Lpn equivalent of GroEL the common heat shock protein Hsp60 of E. coli.  In E. coli, both GroEL and HtpB reside in the cytoplasm, while in Legionella, they are both found extracytoplasmically suggesting a novel secretion mechanism present in Lpn and absent in E. coli.  Building on considerable prior work, Dr. Garduno’s group focus now on how HtbB traverses the cytoplasmic membrane and then how it passes across the outer membrane and eventually into the cytoplasmic vacuole (LCV) of host cells.   The results of these studies are highly dependent on interpretation of cellular localization data from both immunogold studies of ultrathin sections and on enzyme markers of known cellular location.  Finally, the studies employ a range of Dot/Icm mutants (dotA, dotB and IcmE/dotG and their complements).  The authors have gone to great lengths to provide a foundation for measures and counting measures to dissect out cellular location and relative differences between wild type and mutants. It is also fascinating to think that the Dot/Icm secretion system has evolved to recognize highly conserved amino acid sequences in HtpB and GroEL to enable secretion of a protein that is not predicted to reside outside the cytoplasm. The studies are carefully done and the methods used are sufficiently detailed to instill confidence and the authors are careful in interpretation of findings.  Overall, the findings are significant and provide a solid foundation for future research to delineate at a more molecular level.  The manuscript is lengthy, but well written.  Below, I list a few concerns for attention by the authors.

RESPONSE to Comment 1.

We are pleased about the positive general feedback from Reviewer #1, and completely agree with this Reviewer's comment about the length of the manuscript.  We had considered the fact that the manuscript turned out to be quite lengthy, but decided to submit it, and see whether the reviewers would determine that this fact could be a deterrent for readers.   So we are grateful that this reviewer raised this issue, because that directed us to make an extra effort to shorten some sections of the manuscript.  Specifically, we have tried (in the revised version of our manuscript) to shorten the Materials and Methods section, as well as the Results section.

Reviewer #1, Comment 2

The authors might look at some of the methods, such as the Bradford protein determination, and just cite rather than provide details. 

RESPONSE to Comment 2

This is a very good suggestion on how to try to shorten the manuscript.  We have used more citations not only for the Bradford protein determination, but also for other methods, like fractionation, immunoblotting and densitometry, and have provided less methodological details.

Reviewer #1, Comment 3

The semiquantitative methods with immunogold, tables of data etc, an absence of mean and standard deviation should be explained so readers understand that 30 or more cells are evaluated for each data point presented as a mean.  Maybe explain this in the methods section so that readers don’t stumble through this while reading through table legends.

RESPONSE to Comment 3

Yes, we agree that an explanation is required.  The semiquantitative nature of the immunogold labeling data implies that some results were handled in a different (non-traditional) manner.  The approach taken is detailed in reference 24 of the manuscript.  The number of epitopes counted in each bacterial cell compartment was compounded as a single total, divided by 30, and presented in relation to a unit of area (for periplasm or cytoplasm) or a unit of length (for cytoplasmic and periplasmic membranes).  Therefore, we did not obtain the presented results as true means of 30 measurements, but as a single compounded result from 30 sections.  This is now explained in the M&M section of the revised version, and the Tables' legends and footnotes have been shortened to avoid repetition, and make it easier for the reader to follow.

Reviewer #1, Comment 4

Table 3 and elsewhere, with respect to antibody against DsbA2 –to avoid confusion be sure that it is written as DsbA2 and not DsbA as these are distinct proteins.

RESPONSE to Comment 4

This comment has been fully noted, and the nomenclature for this protein has been changed to DsbA2 throughout the revised manuscript,

Reviewer#1, Comment 5

One of the perversities of working with Dot/Icm mutants is that the proteins interact with other members of the Dot/Icm secretion system and with many effector proteins. Ordinarily, the type of mutations used might not matter, but since some mutants might produce truncated proteins that still interact with other proteins, interpretation of results with these mutants might not be straightforward.  Some of this is seen between dotB and delta dotB mutants.  The absence of the protein might be important and the authors comment on this to some extent in the discussion section.  Also, in the case of IcmE/DotG, it is a very large protein and so the use of mutants containing complete deletions of the gene might provide cleaner results.  This is also discussed in the discussion section on DotG. This reviewer worries that it may not be known where the Tn903dIIlacZ insertion is in strain LELA4432.  In this respect, the authors may not be aware of Kpadeh et al (Mol Microbiol. 2015 Mar;95(6):1054-69) where it was found that a complete deletion of dotG had no effect on virulence and rendered the Lp02 strain and JV3559 hyper hemolytic and hyper virulent while strain JV573 expressing a truncated DotG containing an internal deletion was avirulent.  The hyper hemolytic phenotype might suggest HtpB is more efficiently gaining surface location – as mentioned in citation 99.  If it is still true that DotG spans the periplasm or forms a collar as most recently suggested, a truncation might plug the secretion system yielding the avirulent phenotype and a different interpretation with respect to a complete deletion. 

RESPONSE to Comment 5

We are very greatful for both this general observation, and the fact that this Reviewer brought to our attention the Kpadeh et al. 2015 paper.  We have read this paper and agree with the Reviewer's comment that a clean deletion of the dotG gene could have had a different phenotype than the transposon insertion mutant we used.  We have also specified which Lp02 dotG mutant we used in reference 99.  These details have now been added to the Discussion section of the revised manuscript, and also we have emphasized the fact that the analysis of a delta dotL mutant could be more informative than choosing a delta dotG mutant.

Reviewer 2 Report

Comments and Suggestions for Authors

Dear Authors, your work “The passage of chaperonins to extracellular locations in Legionella pneumophila requires a functional Dot/Icm system” dedicated to elucidating the mechanism of transfer of chaperonin HtpB of pathogenic cells into extracellular medium including the cytoplasm of infected cells. It is assumed that the release of the chaperonin into the extracellular space is necessary for the implementation of virulence-related functions that ensure the viability of the pathogen. The work is performed at the modern scientific level and contains a fairly large amount of experimental research using methods of gene engineering, immunology, densitometry and electron microscopy. Despite the fact that the article is quite well written and illustrated, I found several points that should be taken into account in the revised version.

1. Introduction

lines 57 and 61 It would be nice to decipher the abbreviated designation of the cells COH and HtpB-CyaA fusion

2. Materials and Methods

Pages 5 and 6 the reference [43] is not found

Page 7 the reference [51] – line 193, page 8 the reference [55] – line 219, page 9 Table 3 the references [52], [53], [54] - ?

3. Results

Page 13 lines 457 and 472 OM-Surface-decrypt at the first mention

Page 28 The part of text from (line 1036) “Immunogold labeling with αICDH…………” up to (line 1047) “………. the inner membrane to associate with outer membrane” needs appropriate references to experimental or literary data.

Author Response

Reviewer #2, General Comment 1

Dear Authors, your work “The passage of chaperonins to extracellular locations in Legionella pneumophila requires a functional Dot/Icm system” dedicated to elucidating the mechanism of transfer of chaperonin HtpB of pathogenic cells into extracellular medium including the cytoplasm of infected cells. It is assumed that the release of the chaperonin into the extracellular space is necessary for the implementation of virulence-related functions that ensure the viability of the pathogen. The work is performed at the modern scientific level and contains a fairly large amount of experimental research using methods of gene engineering, immunology, densitometry and electron microscopy. Despite the fact that the article is quite well written and illustrated, I found several points that should be taken into account in the revised version.

RESPONSE to General Comment 1

We are pleased with the positive feedback from this Reviewer, and are delighted that both Reviewers of our manuscript agreed that ours is a well written manuscript.

Reviewer #2, Comment 2

  1. Introduction

lines 57 and 61 It would be nice to decipher the abbreviated designation of the cells COH and HtpB-CyaA fusion

RESPONSE to Comment 2

The meaning of CHO (Chinese Hamster Ovary) and CyaA (adenylate cyclase), fused with the C- and N-terminus of HtpB, has been added to the revised version of the manuscript.

Reviewer #2, Comment 3

  1. Materials and Methods

Pages 5 and 6 the reference [43] is not found

Page 7 the reference [51] – line 193, page 8 the reference [55] – line 219, page 9 Table 3 the references [52], [53], [54] - ?

RESPONSE to Comment 3

We have taken extra care to make sure that all the references appear in the correct order of citation in the text, and that all references listed in the References section are cited.  Some references appear in Tables or Figure Legends, or even in the Supplementary Material.  Since the Instructions to Authors indicate that references can be included in the Supplemental Tables and Figures, as long as they are listed in the References section, we included some citations to listed references in the Supplemental Material.  The reader has been alerted to this fact in the revised version of the manuscript.

Reviewer #2, Comment 4

  1. Results

Page 13 lines 457 and 472 OM-Surface-decrypt at the first mention

Page 28 The part of text from (line 1036) “Immunogold labeling with αICDH…………” up to (line 1047) “………. the inner membrane to associate with outer membrane” needs appropriate references to experimental or literary data.

RESPONSE to Comment 4.

Both instances mentioned in this comment have been addressed in the revised version of the manuscript.  The meaning of OM (outer membrane) and "surface|" is now explained, and the references requested (to our experimental data) are now included.